# Comparison of TROPOMI/Sentinel 5 Precursor $NO_2$ observations with ground-based measurements in Helsinki

Iolanda Ialongo[1], Henrik Virta[1], Henk Eskes[2], Jari Hovila[1], and John Douros[2]

[1]Space and Earth Observation Centre, Finnish Meteorological Institute, Helsinki, Finland.
[2]Royal Netherlands Meteorological Institute, De Bilt, Netherlands.

**Correspondence:** Iolanda Ialongo (iolanda.ialongo@fmi.fi)

**Abstract.** We present a comparison between satellite-based TROPOMI (TROPOspheric Monitoring Instrument) $NO_2$ products and ground-based observations in Helsinki (Finland). TROPOMI $NO_2$ total (summed) columns are compared with the measurements performed by the Pandora spectrometer between April and September 2018. The mean relative and absolute bias between the TROPOMI and Pandora $NO_2$ total columns is about $10\,\%$ and $0.12 \times 10^{15}$ molec. $cm^{-2}$, respectively. The dispersion of these differences (estimated as their standard deviation) is $2.2 \times 10^{15}$ molec. $cm^{-2}$. We find high correlation ($r = 0.68$) between satellite- and ground-based data, but also that TROPOMI total columns underestimate ground-based observations for relatively large Pandora $NO_2$ total columns, corresponding to episodes of relatively elevated pollution. This is expected because of the relatively large size of the TROPOMI ground pixel ($3.5 \times 7\,km$) and the a-priori used in the retrieval compared to the relatively small field-of-view of the Pandora instrument. On the other hand, TROPOMI slightly overestimates (within the retrieval uncertainties) relatively small $NO_2$ total columns. Replacing the coarse a-priori $NO_2$ profiles with high-resolution profiles from the CAMS chemical transport model improves the agreement between TROPOMI and Pandora total columns for episodes of $NO_2$ enhancement. When only the low values of NO2 total columns or the whole dataset are taken into account, the mean bias slightly increases. The change in bias remains mostly within the uncertainties.

We also analyse the consistency between satellite-based data and in situ $NO_2$ surface concentrations measured at the Helsinki-Kumpula air quality station (located a few metres from the Pandora spectrometer). We find similar day-to-day variability between TROPOMI, Pandora and in situ measurements, with $NO_2$ enhancements observed during the same days. Both satellite- and ground-based data show a similar weekly cycle, with lower $NO_2$ levels during the weekend compared to the weekdays as a result of reduced emissions from traffic and industrial activities (as expected in urban sites). The TROPOMI $NO_2$ maps reveal also spatial features, such as the main traffic ways and the airport area, as well as the effect of the prevailing south-west wind patterns.

This is one of the first works in which TROPOMI $NO_2$ retrievals are validated against ground-based observations and the results provide an early evaluation of their applicability for monitoring pollution levels in urban sites. Overall, TROPOMI retrievals are valuable to complement the ground-based air quality data (available with high temporal resolution) for describing the spatio-temporal variability of $NO_2$, even in a relatively small city like Helsinki.

# 1 Introduction

Nitrogen oxides ($NO_x = NO + NO_2$) play an important role in tropospheric chemistry, participating in ozone and aerosol production. $NO_x$ are mainly generated by combustion processes from anthropogenic pollution sources (including transportation, energy production and other industrial activities), and they are toxic in high concentrations at the surface (US-EPA, 2019).

The $NO_2$ amount in the atmosphere can be measured using satellite-based instruments. Launched in October 2017, TROPOMI (TROPOspheric Monitoring Instrument), the only payload on-board the European Space Agency's (ESA) Sentinel-5 Precursor (S5P) satellite, is expected to revolutionise the way we monitor air pollution from space because of its unprecedented spatial resolution ($3.5 \times 7$ km at the beginning of the mission and $3.5 \times 5.5$ km since 6 August 2019) and high signal-to-noise ratio. TROPOMI (jointly developed by the Netherlands Space Office and ESA) is designed to retrieve the concentrations of several

atmospheric constituents including ozone, $NO_2$, $SO_2$, CO, $CH_4$, $CH_2O$, aerosol properties as well as surface UV radiation. TROPOMI derives information on atmospheric $NO_2$ concentrations by measuring the solar light back-scattered by the atmosphere and the Earth's surface. Due to its high spatial resolution, TROPOMI observations are particularly suitable to monitoring polluting emission sources at city level. The S5P mission is part of the Space Component of the European Copernicus Earth Observation Programme.

TROPOMI builds on the experience from previous polar orbiting instruments such as the Dutch-Finnish Ozone Monitoring Instrument (OMI), which has been operating on-board NASA's EOS (Earth Observing System) Aura satellite (Levelt et al., 2006) since late 2004. OMI $NO_2$ observations have been used in several air quality applications and the main achievements have been recently summarised by Levelt et al. (2018). The results achieved using OMI $NO_2$ retrievals include estimating top-down polluting emissions, analysing changes in the pollution levels over the period of 13 years, and verifying the success

of environmental policy measures (e.g., Beirle et al., 2011; Castellanos and Boersma, 2012; Streets et al., 2013; Lu et al., 2015; Lamsal et al., 2015; Duncan et al., 2016; Krotkov et al., 2016; Liu et al., 2017). Also, OMI observations have been used for monitoring the $NO_2$ weekly cycle over urban sites (Beirle et al., 2003; Boersma et al., 2009; de Foy et al., 2016). Recently, a reprocessing of the OMI $NO_2$ dataset has become available (Boersma et al., 2018) as deliverable of the European QA4ECV project. Many of the QA4ECV OMI retrieval developments have been incorporated in the TROPOMI $NO_2$ retrieval processor.

Since TROPOMI/S5P is a very recent mission, accurate validation against independent ground-based measurements is needed in order to evaluate the quality of the retrieval. Recently, the Pandonia Global Network (PGN), including a network of ground-based Pandora spectrometers, has been established to provide reference measurements of $NO_2$ total columns for validating satellite-based retrievals. Pandora measures direct sunlight in the ultraviolet-visible spectral range (280–525 nm) and provides $NO_2$ total columns using the direct-sun DOAS (Differential Optical Absorption Spectroscopy) technique (Her-

man et al., 2009). Recently, Zhao et al. (2019) presented a method to derive $NO_2$ total columns from Pandora zenith-sky measurements as well. The TROPOMI/S5P $NO_2$ products are operationally validated by the S5P-MPC-VDAF (S5P - Mission Performance Center - Validation Data Analysis Facility) using the Pandora $NO_2$ total columns from the PGN. The operational validation results are reported every 3 months at the S5P-MPC-VDAF website (http://mpc-vdaf.tropomi.eu/).

Very recently, Griffin et al. (2019) presented first results of the validation of TROPOMI NO$_2$ retrievals over the Canadian oil sands using air-mass factors calculated with the high-resolution GEM-MACH model. They show how the TROPOMI NO$_2$ vertical column densities are highly correlated with ground-based observations and have a negative bias of 15–30 %. In this work, we evaluate the quality of TROPOMI NO$_2$ vertical columns against ground-based observations in the urban site of Helsinki (60.2° N; 24.95° E). Helsinki is a city with about half a million inhabitants, surrounded by a larger urban area (including the city of Espoo in the west and Vantaa in the north-east). Satellite-based NO$_2$ observations from OMI instrument in Helsinki were previously validated by Ialongo et al. (2016), finding that the bias between OMI and Pandora total columns ranges between −30 % and 5 %, depending on the retrieval algorithm and parameters. The improved resolution of TROPOMI retrievals is expected to reduce the effect of spatial averaging compared to OMI, leading to a better agreement with the ground-based Pandora observations that have a relatively narrow field-of-view.

The satellite- and ground-based data used in the analysis are described in Sect. 2. The results of the comparison between TROPOMI NO$_2$ retrievals and ground-based Pandora total columns are shown in Sect. 3. The temporal correlation with in situ NO$_2$ surface concentration measurements and the NO$_2$ weekly cycle are also analysed. Finally, the conclusions are presented in Sect. 4.

## 2 Data and methodology

### 2.1 TROPOMI NO$_2$ observations

TROPOMI is a passive sensing hyperspectral nadir-viewing imager aboard the Sentinel-5 Precursor (S5P) satellite, launched on 13 October 2017. S5P is a near-polar sun-synchronous orbit satellite flying at an altitude of 817 km, with an overpass time at ascending node (LTAN) of 13:30 local time (LT) and a repeat cycle of 17 days (KNMI, 2017). TROPOMI is operated in a non-scanning push broom configuration, with an instantaneous field-of-view of 108° and a measurement period of about 1 second. This results in a swath width of approx. 2600 km, an along-track resolution of 7 km and daily global coverage (KNMI, 2017). TROPOMI's four separate spectrometers measure the ultraviolet (UV), UV-Visible (UV-VIS), near-infrared (NIR) and short-wavelength infrared (SWIR) spectral bands, of which the NIR and SWIR bands are new as compared to its predecessor OMI (Veefkind et al., 2012).

The NO$_2$ columns are derived using TROPOMI's UVIS spectrometer backscattered solar radiation measurements in the 405–465 nm wavelength range (van Geffen et al., 2015, 2019). The swath is divided into 450 individual measurement pixels, which results in a near-nadir resolution of $7 \times 3.5$ km. The total NO$_2$ slant column density is retrieved from the Level 1b UVIS radiance and solar irradiance spectra using the DOAS method (Platt and Stutz, 2008). The species fitted by TROPOMI and their corresponding literature cross sections can be found in van Geffen et al. (2019). Tropospheric and stratospheric slant column densities are separated from the total slant column using a data assimilation system based on the TM5-MP chemical transport model, after which they are converted into vertical column densities using a look-up table of altitude dependent air-mass factors (AMF) and information on the vertical distribution of NO$_2$ from TM5-MP available with a horizontal resolution of $1° \times 1°$ and a time step of 30 minutes (van Geffen et al., 2019; Boersma et al., 2018; Williams et al., 2017).

The instrument, the $NO_2$ retrieval and assimilation scheme, and the data product have been described in detail by Veefkind et al. (2012), KNMI (2017), KNMI (2018), Eskes et al. (2019) and van Geffen et al. (2019).

We used reprocessed (RPRO) TROPOMI $NO_2$ data files, processor version 1.2.2, for the entire study period of 15 April to 30 September 2018. Reprocessed data files are occasionally generated using older sensing data as new processor algorithm versions become available. Version 1.2.x includes retrieval enhancements for high solar zenith angle and snow covered scenes (Eskes et al., 2019), both of which are important for high latitude locations such as Helsinki. The time period of this study did not, however, include any days with snow cover. Additionally, offline (OFFL) and near-real time (NRTI) $NO_2$ products are also available. Offline data files are the main TROPOMI data product and are made available within about two weeks from the sensing time, whereas NRTI files are available within 3 hours of measurement time. NRTI files are generated using forecast TM5-MP data rather than analysis data as with offline and reprocessed files (van Geffen et al., 2019), but the differences between the offline/reprocessed and near-real time products are generally small (Lambert et al., 2019).

The TROPOMI $NO_2$ product used in the comparison was the summed total column, which is the sum of the tropospheric and stratospheric vertical column densities. It was chosen over the total column product, since the latter's sensitivity to the ratio between the stratospheric and tropospheric a-priori columns may lead to substantial systematic retrieval errors. The intermediate step of using data assimilation to first estimate the stratospheric column does remove part of this error. The summed total column product is described by the data provider as the best physical estimate of the $NO_2$ vertical column and recommended for comparison to ground-based total column observations (van Geffen et al., 2019). The precision values of the summed total columns used in the analysis stay within the range $(0.5–4.5) \times 10^{15}$ molec. $cm^{-2}$ (or about 10–50 %). The data before 30 April 2018 were downloaded from the Sentinel-5P Expert Users Data Hub (https://s5pexp.copernicus.eu/dhus) as part of the S5P validation team activities, and starting from this date from the S5P Pre-Operations Data Hub (https://s5phub. copernicus.eu/dhus).

Figure 1 shows the TROPOMI $NO_2$ tropospheric columns over Helsinki averaged over the period 15 April to 30 September 2018. The largest enhancements are visible over the main traffic lanes as well as the Helsinki-Vantaa airport and surrounding area. Overall, the $NO_2$ levels during weekends (Fig. 1, right panel) are smaller than those observed during weekdays (Fig. 1, left panel) by about 30 %. This is typical for urban sites due to the weekly variability of traffic-related emissions, which are relatively higher during working days (from Monday to Friday). We also note that the $NO_2$ spatial distribution shown in Fig. 1 is partially affected by systematic wind patterns, which causes the $NO_2$ levels in the eastern part of the area to become relatively higher than the western part. Fig. S1 in the supplementary material shows the difference between the $NO_2$ tropospheric columns (normalised to the maximum value in the area) for all wind and low wind speed (less than $3\,m\,s^{-1}$) conditions. The pixels in red and blue in Fig. S1 indicate the area where the $NO_2$ levels are relatively higher or lower, respectively, due to the wind patterns. This is related to the prevailing wind directions from south-west over the Helsinki capital region.

Since the retrieval of TROPOMI vertical column densities (VCDs) is sensitive to the a-priori estimate of the $NO_2$ profile shape, the accuracy of the VCDs may be improved by using a-priori profiles from a chemical transport model (CTM) with a higher resolution than the $1° \times 1°$ of TM5-MP (Williams et al., 2017). The air-mass factor (AMF) can be recomputed using

an alternative a-priori $NO_2$ profile, resulting in a new retrieval of the tropospheric $NO_2$ column as described by Eskes et al. (2019).

In order to analyse their impact on the comparison, below 3 km altitude we used $NO_2$ profiles from the CAMS regional ENSEMBLE model (Météo-France, 2016; Marécal et al., 2015) as an alternative to the TM5-MP profiles. The CAMS regional ENSEMBLE is a median of seven European CTMs, and the data are provided on a regular $0.1° \times 0.1°$ grid over Europe on 8 vertical levels up to 5 km altitude. In addition, the CAMS global model was used to generate the profiles above 3 km altitude with the assumption that this model gives a more reliable description of NOx in the free troposphere. Data for CAMS global are provided on a regular $0.4° \times 0.4°$ grid on 60 model levels reaching up to 0.1 hPa (Flemming et al., 2015). In particular, we used the ratios between TROPOMI tropospheric air-mass factors derived using the hybrid CAMS regional/global a-priori profile (henceforth "CAMS a-priori") and the TM5-MP a-priori profile (see Sect 2.3). These ratios were derived on the regular CAMS $0.1° \times 0.1°$ grid for the period 30 April to 30 September 2018.

In order to minimize representativeness errors during the comparison, certain considerations were taken into account so that the fields could be correctly sampled in space and time. Horizontally, all available gridded data were interpolated to the CAMS regional, $0.1° \times 0.1°$ grid. Source grids in this process were either the TROPOMI native grid which is different for each orbit, the CAMS global grid or the TM5-MP grid. Horizontal interpolation of retrieval columns was realized by means of a weighted average of all individual columns within a target grid cell. Intensive variables (e.g. temperatures, pressures, averaging kernels, the tropopause layer index etc.) were interpolated horizontally using bilinear regridding. Modelled fields were also interpolated in time, based on the satellite overpass time over Central Europe. All vertical levels of source data were linearly interpolated to the TM5-MP vertical levels and all subsequent integrations to columns were performed based on those levels. Pressures at each of those levels were calculated based on the surface pressure and the hybrid coefficients included in the TROPOMI product, which originate in TM5-MP. For the column integrations, all concentrations were converted to densities based on temperature and pressure profiles provided by TM5-MP.

## 2.2 Ground-based $NO_2$ observations

The $NO_2$ total columns measured by the ground-based Pandora instrument #105 located in the district of Kumpula, Helsinki, Finland ($60.20°$ N, $24.96°$ E), are compared to the TROPOMI $NO_2$ retrievals. The Pandora system is composed of a spectrometer connected by a fibre optic cable to a sensor head with $1.6°$ FOV (field-of-view). A sun-tracking device allows the optical head to point at the centre of the Sun with a precision of $0.013°$ (Herman et al., 2009). Pandora performs direct-sun measurements in the UV-VIS spectral range (280–525 nm) and provides $NO_2$ total vertical column densities, among other products.

The $NO_2$ total column retrieval is based on the DOAS spectral fitting technique (e.g., Cede et al., 2006), with $NO_2$ and $O_3$ being the trace gases fitted. The algorithm derives the relative $NO_2$ slant column densities (SCDs) from the 400–440 nm spectral band and converts them to absolute SCDs using a statistically estimated reference spectrum obtained using the Minimum-Amount Langley-Extrapolation method (MLE) (Herman et al., 2009).

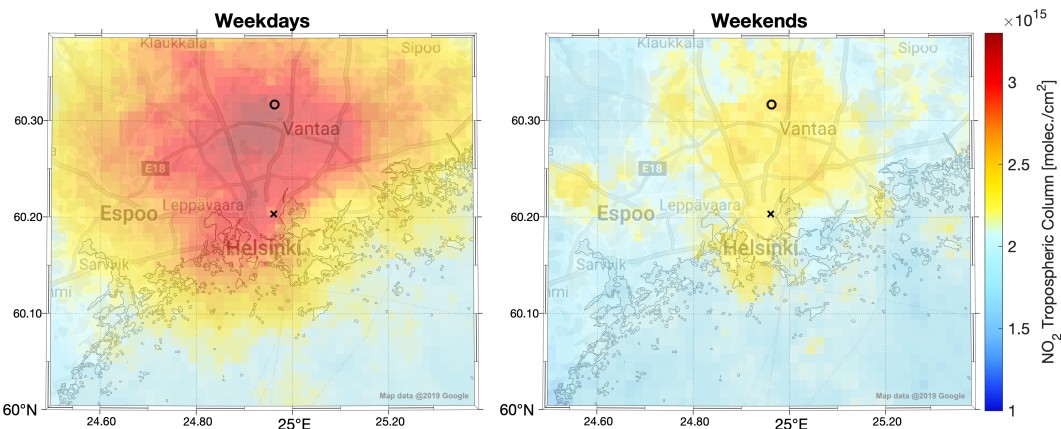

**Figure 1.** Average TROPOMI $NO_2$ tropospheric columns over Helsinki during the period 15 April to 30 September 2018. The left and right panels correspond to weekdays and weekends, respectively. The data have been binned and averaged to a 1 km resolution grid. The locations of the Kumpula ground-based station and the Helsinki-Vantaa airport are shown with a black cross and circle, respectively.

The Pandora SCD retrieval employs a temperature correction to the cross-sections used in the spectral fitting procedure based on modelled monthly average $NO_2$ and temperature profiles and high-resolution temperature-dependent cross sections by Vandaele et al. (1998) for $NO_2$ and Serdyuchenko et al. (2014) for $O_3$ (as in the TROPOMI retrieval). We note that, while TROPOMI uses the ECMWF operational model as its source for atmospheric temperature profiles (van Geffen et al., 2019), Pandora uses a precalculated atmospheric temperature for a typical $NO_2$ profile (Cede, 2019). Due to the nature of direct-sun measurements no Ring effect correction is needed for Pandora (Herman et al., 2009).

The $NO_2$ columns are available about every 1.5 minutes. The full description of the Pandora instrument and the algorithm for the inversion methodology has been presented by Herman et al. (2009). The nominal clear-sky precision of the Pandora $NO_2$ total column retrievals is in the order of $0.3 \times 10^{15}$ molec. $cm^{-2}$ with an accuracy of about $\pm 1.3 \times 10^{15}$ molec. $cm^{-2}$. The accuracy depends on the uncertainties in the MLE-calculated reference spectrum, difference between the actual and assumed atmospheric temperature profiles, and uncertainties in the laboratory-determined absorption cross sections (Herman et al., 2009). At typical Helsinki concentrations ($6 \times 10^{15}$ molec. $cm^{-2}$) and AMF values (2.0) most of the systematic errors are due to uncertainties in the reference spectrum (Sect. 3.3 in Herman et al. (2009)). Pandora #105 is part of the Pandonia global network and the observations used in this paper were processed following the Pandonia procedure and distributed at http://pandonia.net/data/.

The $NO_2$ surface concentrations available from the Kumpula, Helsinki, air quality (AQ) station were used in order to analyse the temporal correspondence between surface $NO_2$ concentrations and TROPOMI vertical columns. This station, also known as the SMEAR III station (Järvi et al., 2009), is located close to the Pandora instrument (about 100 m distance), and is classified as a semi-urban site. Nitrogen oxides are measured using a chemiluminescence-based analyser (HORIBA APNA-360, Kato and Yoneda, 1997). $NO_x$ and NO measurements from the station are available from the SmartSMEAR online service in intervals of one minute and in units of ppb (https://avaa.tdata.fi/web/smart), while $NO_2$ measurements are available from the FMI (Finnish

Meteorological Institute) measurement database as hourly averaged concentrations in units of µg m$^{-3}$ (no open access). The air quality data were linearly interpolated to TROPOMI overpass times when compared with collocated Pandora and TROPOMI data. The middle of the one hour averaging period was used as the time stamp for the AQ measurements, as it was found that this resulted in the best correlation with collocated Pandora measurements.

## 2.3 Methodology

We evaluate the agreement between TROPOMI and Pandora NO$_2$ vertical column densities by calculating the mean absolute difference (MD), the mean relative difference (MRD), the dispersion (i.e., the standard deviation) of the differences ($\sigma$), the correlation coefficient (r), and the slopes of ordinary least squares and York linear regression fits for the measurements. The MD is defined as the average difference between the TROPOMI and Pandora VCDs in Eq. (1), whereas the MRD is the average of these differences when normalised with Pandora's VCD (Eq. (2)).

$$\text{MD} = \frac{1}{n}\sum_{i=1}^{n}\left(\text{VCD}_{\text{TROPOMI},i} - \text{VCD}_{\text{Pandora},i}\right) \tag{1}$$

$$\text{MRD} = 100\,\% \times \frac{1}{n}\sum_{i=1}^{n}\frac{\text{VCD}_{\text{TROPOMI},i} - \text{VCD}_{\text{Pandora},i}}{\text{VCD}_{\text{Pandora},i}} \tag{2}$$

A positive MD or MRD is thus an indication of TROPOMI overestimation, and negative an indication of TROPOMI underestimation. The York linear regression (York et al., 2004) is used alongside the traditional least squares linear regression, since it has been shown to be appropriate in situations where the two sets of data have different levels of uncertainty (Wu and Yu, 2018). We also analyse weekdays and weekends separately, and the results are presented in Sect. 3.

Both TROPOMI and Pandora data were separately filtered according to a set of quality assurance criteria, after which the remaining temporally co-located measurements were compared with each other. For TROPOMI, only measurements with a data quality value QA >0.75 are used, which disqualifies scenes with a cloud radiance fraction >0.5, some scenes covered by snow or ice, and scenes that have been determined to include errors or problematic retrievals. Further details on the QA value are provided in the appendices of van Geffen et al. (2019). Only TROPOMI pixels including the Helsinki Pandora station were considered for the comparison. Also, only Pandora retrievals with data quality flag value of 0, 1, 10 or 11, corresponding to so-called assured and not-assured high or medium quality data (Cede, 2019), were taken into account. Pandora measurements within 10 minutes of TROPOMI overpass were averaged to get the Pandora-component of the validation data pairs. Wind speed data (average from the four lowest pressure levels: 925, 950, 975 and 1000 hPa) available from the European Centre for Medium-Range Weather Forecasts (ECMWF) as part of the ERA5 reanalysis product (https://cds.climate.copernicus.eu) were associated with each data pair in order to quantify the effect of advection on the NO$_2$ concentrations. The wind data were linearly interpolated to the Helsinki Pandora station's coordinates and the overpass time of each TROPOMI pixel used in the comparison.

Furthermore, we analyse the effect of the co-location choices on the MD, MRD, standard deviation of the differences $\sigma$ and correlation coefficient by varying both the maximum distance from the ground-based station and the averaging time interval for Pandora measurements around the S5P overpass time. The results are presented in Sect. 3. When calculating these values

for increasing maximum distances, we also required that in all cases the TROPOMI pixel above the station had to have a valid measurement fulfilling our quality criteria.

The effect of using high-resolution CAMS a-priori $NO_2$ profiles instead of TM5-MP (as used in the standard product) in the calculation of TROPOMI VCDs was analysed by calculating an alternative summed column using the ratio (R) between the tropospheric air-mass factors derived using CAMS and TM5-MP a-priori profiles, computed on the CAMS-regional grid with 0.1° resolution (see Sect. 2.1). For each available orbit we used the value of R in the CAMS grid pixel that included the Pandora station. The new summed column, derived using the CAMS a-priori profile, was then calculated from the tropospheric and stratospheric $NO_2$ VCDs of the standard L2 product as

$$VCD_{\text{summed, CAMS}} = R \times VCD_{\text{tropos, TM5-MP}} + VCD_{\text{stratos, TM5-MP}} . \tag{3}$$

The stratospheric columns from TM5-MP (as in the standard product) are used in the calculation of the new summed columns, because at the moment CAMS global does not include detailed stratospheric chemistry nor accurate $NO_2$ profile information in the stratosphere. The new TROPOMI-CAMS summed columns calculated using Eq. (3) were then also compared to the Pandora total columns, and the results are presented in Table 2 and Fig. 8. Apart from these two instances, all tables and figures in this paper use standard TROPOMI data products (i.e. based on TM5-MP a-priori profiles).

# 3   Results

Figure 2 shows the time series of the $NO_2$ measurements used in the analysis, covering the period April to September 2018. The Pandora $NO_2$ total columns are shown in their original time resolution (blue dots) as well as averaged 10 minutes around the S5P overpass (red dots). The latter are used in the quantitative comparison to the TROPOMI $NO_2$ summed columns (yellow diamonds). The hourly $NO_2$ surface concentrations measured at Kumpula AQ station are also shown on the right hand y-axis (black line). The Pandora total columns and the surface concentrations show similar peaks and day-to-day variability (blue dots and black line, respectively), which shows how the Pandora observations are sensitive to the changes in the $NO_2$ levels occurring at the surface. We note that the collocated TROPOMI and Pandora vertical columns (yellow diamonds and red dots, respectively, in Fig. 2) also mostly follow the same day-to-day variability. The largest differences between TROPOMI and Pandora vertical columns, with TROPOMI smaller than Pandora, correspond to relatively high $NO_2$ enhancements measured at the surface (black line in Fig. 2). This is expected, as the comparatively large size of the TROPOMI pixels leads to greater spatial averaging compared to the Pandora field-of-view.

In order to further compare satellite- and ground-based collocated observations, Fig. 3 shows the scatter plot between Pandora and TROPOMI total columns from the overpasses presented in Fig. 2. The filled dots correspond to weekdays while the empty circles to the weekends. The colour indicates the corresponding wind speed. The weekend overpasses fall mostly into the bottom-left area of the scatter plot, corresponding to relatively small $NO_2$ total columns from both Pandora and TROPOMI retrievals. This is expected due to the $NO_2$ weekly cycle over urban sites, i.e. reduced polluting emissions from traffic during the weekend compared to the weekdays. Furthermore, the overpasses corresponding to high wind speed values (green-yellow

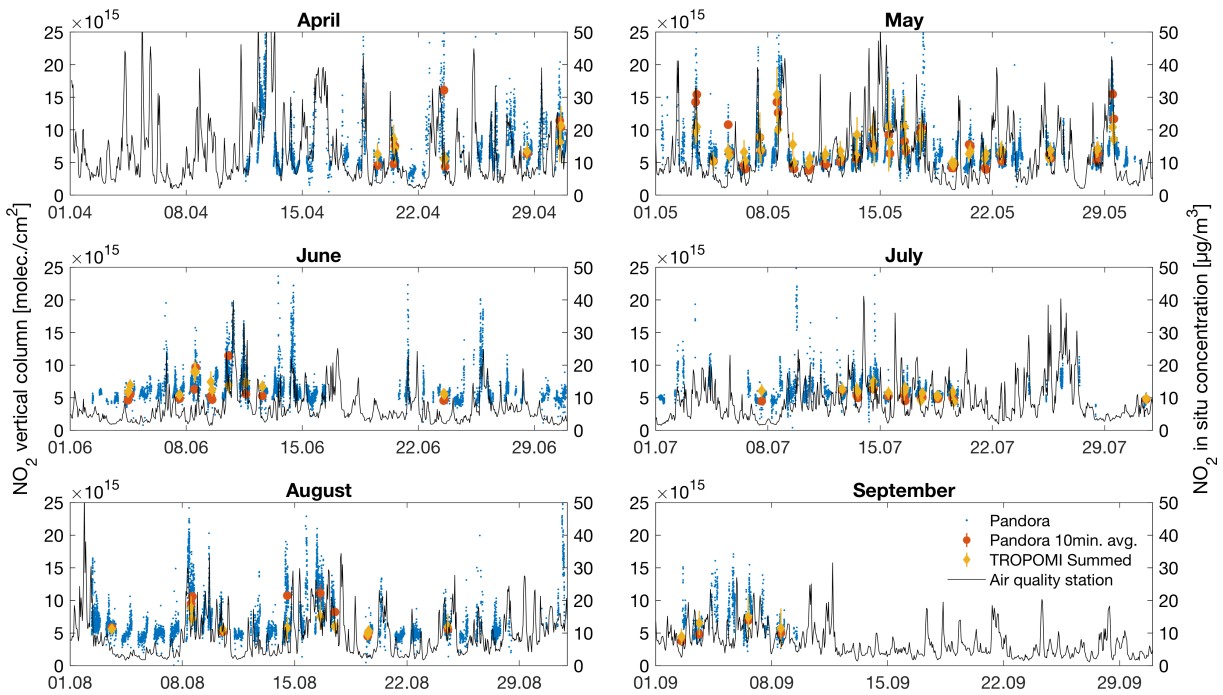

**Figure 2.** Time series of co-located Pandora total and TROPOMI summed $NO_2$ columns during the period 15 April to 30 September 2018. Blue dots are all the available Pandora observations; red dots are the Pandora observations averaged 10 minutes before and after S5P's overpass (with standard errors of the mean as error bars); yellow diamonds are TROPOMI summed columns of the pixels including the ground-based Pandora station (with retrieval precisions as error bars). The black line (right y-axis) indicates the $NO_2$ surface concentrations from the in situ measurements at the Kumpula AQ station.

colours in Fig. 3) also fall into the bottom-left area of the scatter plot. In these cases, the dilution by the wind acts to reduce the $NO_2$ levels. Overall, the data points are quite close to the one-to-one line, except for some cases with elevated $NO_2$ total columns measured by Pandora. These cases correspond to $NO_2$ enhancements with small wind speed (below $3\,\mathrm{m\,s^{-1}}$), when the spatial dilution effect of TROPOMI's ground footprint as compared to Pandora's narrow field-of-view is especially pronounced.

5    Table 1 summarises the results of the comparison between TROPOMI and Pandora in terms of mean relative difference (MRD), mean difference (MD), standard deviation of the difference ($\sigma$), correlation coefficient (r), slopes of linear and York regression fits, and number of overpasses (n). The overall MRD and MD values are $(9.9 \pm 2.6)\,\%$ and $(0.12 \pm 0.22) \times 10^{15}\,\mathrm{molec.\,cm^{-2}}$, respectively, meaning that on average TROPOMI slightly overestimates the $NO_2$ total columns. The dispersion of these absolute differences, calculated as their standard deviation, is $2.2 \times 10^{15}\,\mathrm{molec.\,cm^{-2}}$. The correlation coefficient is high ($r = 0.68$).

10   When considering only weekdays, the MD and MRD values become slightly smaller (MRD=$(9.0 \pm 3.3)\,\%$) but the change remains within the uncertainties. This is expected, as weekday observations contain a number of collocations where the difference between TROPOMI and Pandora vertical columns is exceedingly negative (Fig. 3), corresponding to $NO_2$ enhancements measured by Pandora. Correspondingly, the MRD and MD values for the weekend (typically associated with lower $NO_2$ lev-

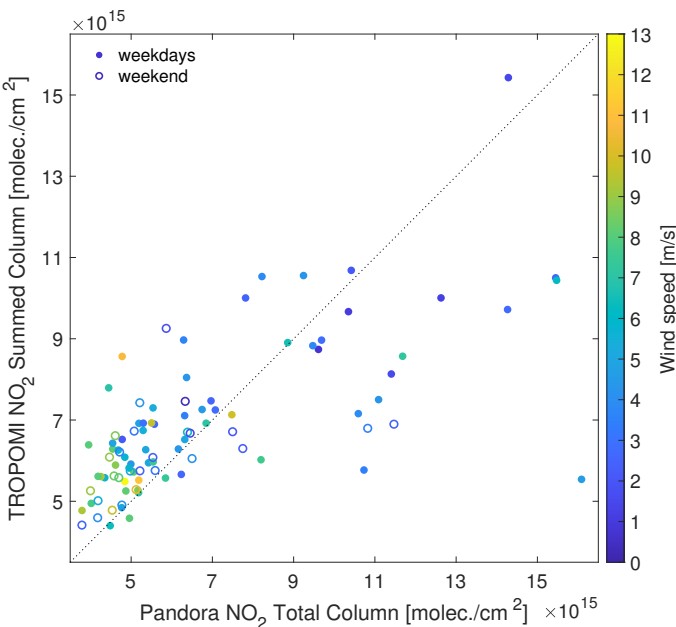

**Figure 3.** Scatter plot between Pandora and TROPOMI vertical columns. The filled dots correspond to weekdays while the empty circles to the weekends. The colour indicates the wind speed interpolated at the overpass time. The 1:1 line is plotted as dotted line.

**Table 1.** Statistics of the comparison between TROPOMI and Pandora $NO_2$ total columns. The uncertainties are the corresponding standard errors of the mean. The uncertainty estimates used in the York fit are pixel-specific precisions for TROPOMI (included in the data product), and standard errors of the mean for Pandora as calculated for the set of measurements within 10 minutes of the S5P overpass.

|  | MRD[a] | MD[b] | $\sigma^c$ | r[d] | $\text{slope}^e_{LS}$ | $\text{slope}^f_Y$ | n[g] |
|---|---|---|---|---|---|---|---|
| all data | $9.9 \pm 2.6$ | $0.12 \pm 0.22$ | 2.2 | 0.68 | 0.42 | 0.36 | 94 |
| Pandora HQ[h] | $10.1 \pm 3.6$ | $0.08 \pm 0.32$ | 2.4 | 0.66 | 0.41 | 0.33 | 56 |
| weekdays | $9.0 \pm 3.3$ | $0.02 \pm 0.29$ | 2.3 | 0.68 | 0.42 | 0.37 | 67 |
| weekends | $12.1 \pm 4.4$ | $0.38 \pm 0.32$ | 1.7 | 0.46 | 0.26 | 0.32 | 27 |
| Pandora high[i] | $-28.1 \pm 4.8$ | $-3.60 \pm 0.70$ | 2.7 | 0.31 | 0.38 | 0.19 | 15 |
| Pandora low[j] | $17.1 \pm 2.2$ | $0.83 \pm 0.12$ | 1.1 | 0.72 | 0.69 | 0.61 | 79 |

[a] Mean Relative Difference [%]; [b] Mean Difference [$\times 10^{15}$ molec. cm$^{-2}$]; [c] Standard deviation of absolute bias [$\times 10^{15}$ molec. cm$^{-2}$]; [d] Correlation coefficient; [e] Least squares linear fit slope; [f] York linear fit slope; [g] Number of collocations; [h] High quality Pandora observations (QA value 0 or 10); [i] Pandora $NO_2$ total columns $\geq 10 \times 10^{15}$ molec. cm$^{-2}$; [j] Pandora $NO_2$ total columns $< 10 \times 10^{15}$ molec. cm$^{-2}$.

els) are larger. When taking into account only overpasses with Pandora $NO_2$ columns larger than $10 \times 10^{15}$ molec. cm$^{-2}$, the bias becomes exceedingly negative (about $-28\%$ or $(-3.60 \pm 0.70) \times 10^{15}$ molec. cm$^{-2}$), meaning that TROPOMI underestimates the $NO_2$ total columns when $NO_2$ enhancements occur. When considering overpasses below that threshold, the bias is

positive (about 17 %). These two effects partially cancel each other when the data set is considered as a whole. Figure S2 in the Supplement illustrates in more details how the bias changes from positive (about $10^{15}$ molec. cm$^{-2}$) to negative (almost $-4 \times 10^{15}$ molec. cm$^{-2}$) for increasing values of Pandora NO$_2$ total columns. The standard deviation of differences and the correlation coefficient are smaller for weekend overpasses and low Pandora NO$_2$ total columns compared to weekdays and high Pandora NO$_2$ total columns. We also note that taking into account only Pandora retrievals with the highest quality flagging (0 or 10) does not have a substantial effect on the results of the comparison (second row of Table 1), but it reduces the amount of data available for the comparison by about 40 % (as compared to the case where also medium quality data are included).

Figure 4 shows how the choice of the overpass criteria affects the calculated MD value (a similar plot for the MRD is shown in Fig. S3 of the Supplement). In the analysis presented so far we have included only measurements from those TROPOMI pixels which include the Pandora ground-based station. It is also possible to average the contribution from all those pixels which fall within a certain distance from the station. Figure 4 (upper panel) shows how the MD gradually shifts towards negative values (from about $0.1 \times 10^{15}$ to $-0.5 \times 10^{15}$ molec. cm$^{-2}$) when the radius increases from 5 to 30 km. This suggests that averaging over a larger area causes the resulting TROPOMI vertical columns (used in the comparison) to become smaller than those obtained from the single overlaying pixel because of the inhomogeneous spatial distribution of NO$_2$, so that the mean concentrations decrease with increasing distance. The MD (and MRD) value for the overlaying pixel criterion is very similar to the value obtained for the distance of 5 km, even if the number of collocations is not exactly the same. Also, the correlation coefficient value decreases and the standard deviation of the differences increases while the radius increases (upper panels in Fig. S4 and S5, respectively, in the supplement).

Similarly, Fig. 4 (lower panel) shows how the MD value changes when the Pandora observations are averaged over an increasing time range, from 5 to 55 min around the overpass time of the satellite. The MD value increases with increasing temporal averaging interval by about $0.3 \times 10^{15}$ molec. cm$^{-2}$ (2 percentage points). Averaging over an increasing time range generally slightly reduces the Pandora total column values used in the comparison with TROPOMI, making the MD more positive. The correlation coefficient value decreases until 20 km radius while slightly increasing for larger radius values while the standard deviation of the differences behaves in the opposite way (lower panels in Fig. S4 and S5, respectively, in the supplement).

Figure S6 in the supplement includes the absolute differences between TROPOMI and Pandora NO$_2$ total columns as a function of TROPOMI SZA (solar zenith angle) and CRF (cloud radiance fraction) (upper and lower panel, respectively) within the range of values allowed after the TROPOMI data screening (QA value >0.75). The differences between satellite- and ground-based retrievals for SZA above 45° are generally larger (between $-3$ and $1 \times 10^{15}$ molec. cm$^{-2}$) than for smaller values (0 to $1 \times 10^{15}$ molec. cm$^{-2}$). Similarly, larger CRF values correspond to larger (positive or negative) absolute differences.

Since S5P has often two valid overpasses per day at the latitude of Helsinki (60° N), it is possible to study the NO$_2$ daily variability in the time range between about 12 and 15 LT. The S5P overpass time typically corresponds to the NO$_2$ daily local minimum (between the morning and afternoon peaks due to commuter traffic), observed for example in the NO$_2$ surface concentration measurements from Kumpula AQ site (Fig. S7). Figure 5 (upper panel) shows TROPOMI and Pandora NO$_2$ total columns as a function of the time of the day between 12 and 15 LT. Both datasets show an enhancement around 13:30 LT and

lower NO$_2$ levels before and after. The relative differences between TROPOMI and Pandora NO$_2$ total columns do not show a clear dependence on the time of the day (Fig. 5, lower panel), but the dispersion (standard deviation of the relative differences) is larger before 13:30 LT (about 30 %) than afterwards (21 %). Increasing time of the day also corresponds to increasing pixel number (colour of the filled dots in Fig. 5, lower panel), with the first overpass of the day corresponding to the left side of the swath (smaller pixel numbers) and the second overpass to the right side (higher pixels number). No clear dependence between the relative differences and the pixel size (larger at the edges and smaller in the centre of the swath) was observed.

In order to better compare the temporal variability of the NO$_2$ vertical columns and surface concentrations, we employ a simple empirical conversion based on the linear regression between Pandora vertical columns and surface concentrations measured at the Kumpula AQ site, at the satellite overpass time (Fig. 6, left panel). From the results of the linear fit (showing high correlation, r = 0.74), we convert the surface concentrations into total columns and compare the results to the TROPOMI and Pandora time series (Fig. 6, right panel). We note how the three datasets show a very similar temporal variability, with NO$_2$ peaks occurring during the same days. We particularly note NO$_2$ enhancements in May and during the first half of August.

We also analyse the NO$_2$ weekly cycle as seen from the different datasets. Figure 7 shows the Pandora NO$_2$ total columns, TROPOMI summed and tropospheric columns and surface concentrations at the Kumpula air quality station as a function of the day of the week. The data are normalised by the corresponding weekly mean value. We note that all datasets show smaller values on Saturdays and Sundays, as expected from the weekly cycle of NO$_x$ emissions typical of urban sites. The NO$_2$ surface concentrations show about 30–50 % smaller values in the weekend compared to the weekly average, while TROPOMI tropospheric columns are about 20–30 % lower. Pandora and TROPOMI summed NO$_2$ vertical columns are also lower in the weekends (compared to the corresponding weekly means), but only by about 10–20 %. This is because no weekend effect is expected in the stratospheric fraction of the NO$_2$ column. Surface NO$_2$ concentration measurements can be expected to show a larger difference between weekend and weekdays due to their greater sensitivity to changes in polluting emissions at the surface (especially from traffic in the urban environment). The results are consistent with those found using nine years of OMI NO$_2$ observations in Helsinki (Ialongo et al., 2016).

Finally, we evaluate the effect of using the NO$_2$ a-priori profiles derived from the high-resolution CAMS regional ENSEM-BLE model, instead of profiles from the TM5-MP CTM as used in TROPOMI's standard product, in the calculation of NO$_2$ vertical column densities. Figure 8 shows the comparison of the standard product summed columns and the summed columns derived using the CAMS a-priori profiles, calculated as described in Sect. 2.3, to the Pandora total columns (analogously to Fig. 3). Only those overpasses (n=75) for which both a-priori summed columns were available were included in the comparison. The statistics are presented in Table 2 and the corresponding time series in Fig. S8 of the supplement. The comparison shows that the largest differences between the two summed columns are mostly found in cases of relatively high concentrations. In these cases, the use of CAMS profiles generally increases the TROPOMI summed columns and reduces the difference between TROPOMI and Pandora (from $(-28.5 \pm 3.3)$ % for TM5-MP to $(-23.7 \pm 3.5)$ % for CAMS). On the other hand, in cases of low concentrations, where TROPOMI tends to overestimate the VCDs compared to Pandora, the use of CAMS a-priori profiles slightly increases the positive bias (from $(16.9 \pm 2.3)$ % for TM5-MP to $(19.1 \pm 2.3)$ % for CAMS). Because the largest improvement is achieved for relatively high concentrations and negative biases becoming less negative, the overall MRD value

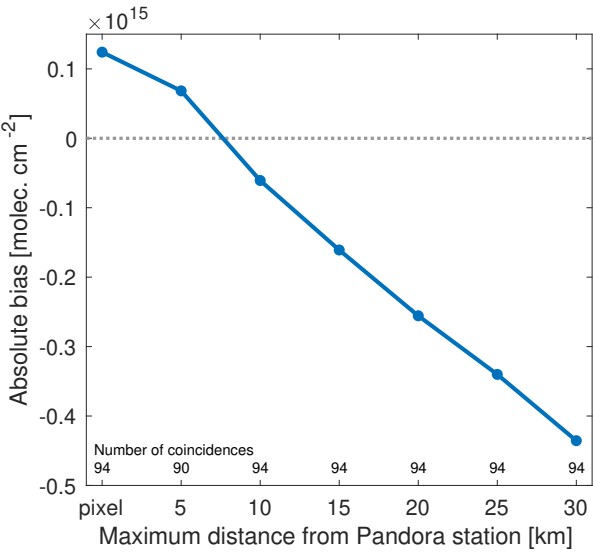

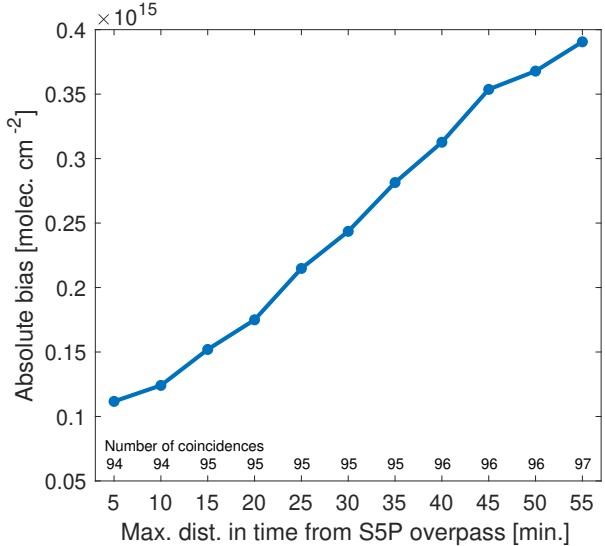

**Figure 4.** Mean absolute difference between TROPOMI summed and Pandora total columns as a function of the maximum distance between the centre of the pixel and the ground-based station (upper panel), and as a function of the maximum time difference from TROPOMI overpass (lower panel). The number of coincidences for different collocation criteria are shown above the x-axis. Note that in the upper panel we also require that the TROPOMI pixel above Pandora station contains a valid measurement (QA value >0.75). Thus the number of coincidences does not increase with distance.

increases from 11.5 % to 14 % (Table 2). According to a two-sided $t$-test, the differences of the two mean absolute biases (MD)

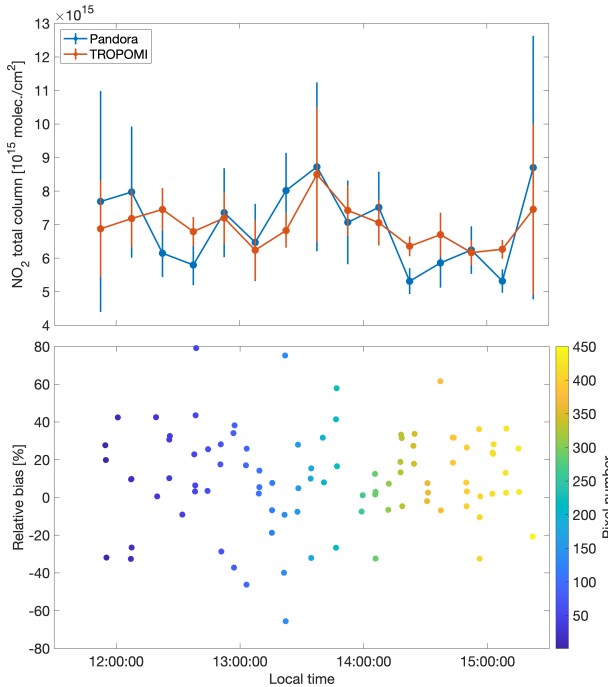

**Figure 5.** Upper panel: TROPOMI NO$_2$ summed columns and Pandora total columns as a function of the time of the day between about 12 and 15 LT. The error bars are the standard errors of the mean. Lower panel: Relative difference between TROPOMI summed columns and Pandora total columns as a function of the time of the day. Filled colours correspond to the TROPOMI pixel number.

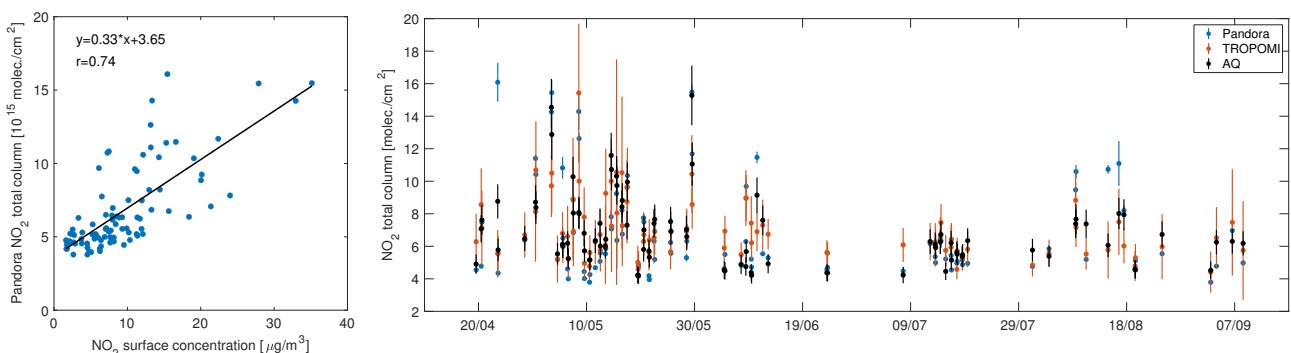

**Figure 6.** Right panel: Time series of NO$_2$ total columns from Pandora (blue), TROPOMI (red) and Kumpula AQ station (black) at the satellite overpass time. The surface concentrations are empirically converted to total columns using the results of the linear regression between Pandora total columns and surface concentration data (left panel).

in Table 2 are statistically significant only at the 52 % significance level. Thus, on average, the use of CAMS profiles does not significantly improve the agreement with Pandora observations.

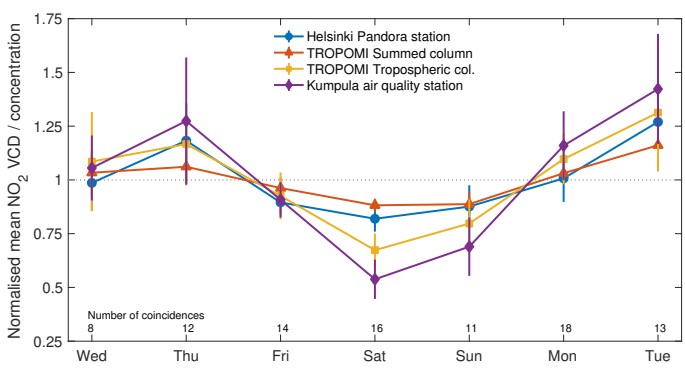

**Figure 7.** NO$_2$ weekly cycle in Helsinki. The average of temporally co-located values for each day of the week for Pandora total columns (blue line), TROPOMI summed and tropospheric columns (red and yellow line, respectively), and surface concentrations as measured at the Kumpula AQ station (purple line) are shown. Error bars represent corresponding standard errors of the mean. All values have been normalised by the weekly mean of each data set.

For this smaller subset of 75 co-locations with Pandora the correlation between TM5-MP summed columns and Pandora is 0.74 and the slope of a least squares linear fit is 0.45. Using the CAMS profiles improves the agreement with Pandora in terms of correlation and slope, with their values increasing to 0.80 and 0.52, respectively. This improvement is more evident for high values of the Pandora NO$_2$ total columns with the correlation and the linear slope increasing by 0.1 and 0.27, respectively, from TM5-MP to CAMS (Table 2). The time series in Fig. S8 of the supplement further show how using the high-resolution CAMS profiles increases the TROPOMI tropospheric columns so that the summed columns (yellow dots) become closer to Pandora's peak values (blue dots), corresponding to episodes of NO$_2$ enhancement, but that overall the difference between the summed columns obtained using TM5-MP and CAMS remains mostly within the uncertainties of the TROPOMI NO$_2$ retrieval.

## 4   Conclusions

We showed the results of the comparison between satellite-based TROPOMI/S5P NO$_2$ products and ground-based observations at a medium-sized urban site, Helsinki (Finland). We find that the differences between the total columns derived from the TROPOMI and Pandora instruments are on average around 10 % (or $0.12 \times 10^{15}$ molec. cm$^{-2}$), which is smaller than the precision of the TROPOMI summed columns used in this study (10–50 %) and well below the requirements for TROPOMI observations (25–50 % for the NO$_2$ tropospheric column and $<10$ % for the stratospheric column; ESA, 2017). The day-to-day and weekly NO$_2$ variability (typical of urban sites) is reproduced well by the TROPOMI retrievals, similarly to Pandora and in situ surface observations from the local air quality station. This confirms that the satellite-based TROPOMI/S5P NO$_2$ retrievals are sensitive to changes in air pollution levels occurring at the surface.

In general, we find that TROPOMI NO$_2$ summed columns are smaller than Pandora total columns for relatively high concentrations, while low values are overestimated. This is partly due to the low resolution of the TM5-MP profile shapes used

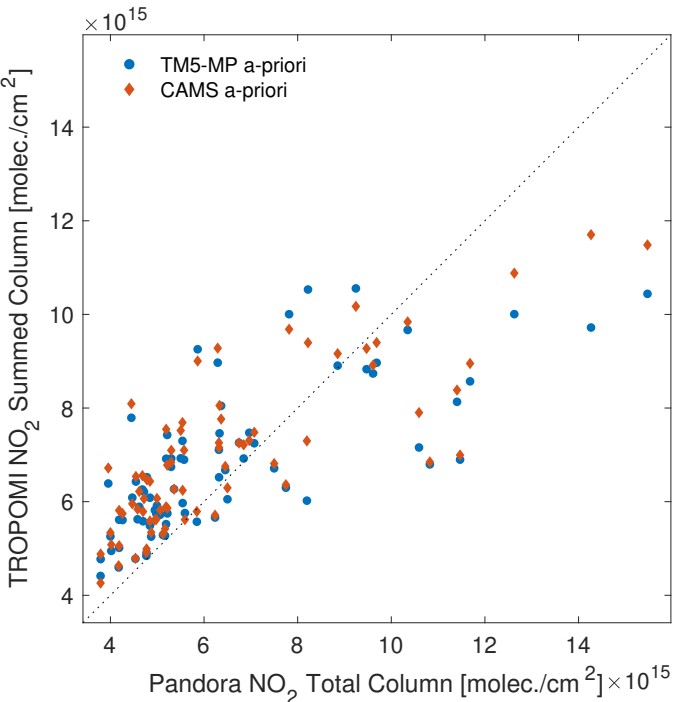

**Figure 8.** Scatter plot between Pandora and TROPOMI summed columns derived using CAMS regional and TM5-MP a-priori NO$_2$ profiles (blue dots and red diamonds, respectively). The comparison includes only those overpasses for which both summed columns were available at the same time during the time interval 30 April to 30 September 2018. The 1:1 line is plotted as dotted line.

**Table 2.** Statistics of the comparisons between TROPOMI summed columns calculated using two different a-priori NO$_2$ profiles (TM5-MP and CAMS regional) and Pandora total columns during 30 April to 30 September 2018. The uncertainties are given as standard errors of the mean.

|  | MRD$^a$ | MD$^b$ | $\sigma^c$ | r$^d$ | slope$_{LS}^e$ | n$^f$ |
|---|---|---|---|---|---|---|
| TM5-MP | $11.5 \pm 2.7$ | $0.31 \pm 0.20$ | 1.8 | 0.74 | 0.45 | 75 |
| CAMS | $14.0 \pm 2.6$ | $0.49 \pm 0.18$ | 1.6 | 0.80 | 0.52 | 75 |
| TM5-MP high$^g$ | $-28.5 \pm 3.3$ | $-3.48 \pm 0.44$ | 1.3 | 0.67 | 0.55 | 9 |
| CAMS high$^g$ | $-23.7 \pm 3.5$ | $-2.86 \pm 0.41$ | 1.2 | 0.77 | 0.82 | 9 |
| TM5-MP low$^h$ | $16.9 \pm 2.3$ | $0.83 \pm 0.13$ | 1.0 | 0.75 | 0.71 | 66 |
| CAMS low$^h$ | $19.1 \pm 2.3$ | $0.95 \pm 0.12$ | 0.97 | 0.78 | 0.72 | 66 |

$^a$Mean Relative Difference [%]; $^b$Mean Difference [$\times 10^{15}$ molec. cm$^{-2}$]; $^c$Standard deviation of absolute bias [$\times 10^{15}$ molec. cm$^{-2}$]; $^d$Correlation coefficient; $^e$Least squares linear fit slope; $^f$Number of collocations; $^g$Pandora NO$_2$ total columns $\geq 10 \times 10^{15}$ molec. cm$^{-2}$; $^h$Pandora NO$_2$ total columns $< 10 \times 10^{15}$ molec. cm$^{-2}$.

to compute the tropospheric air-mass factors and thus the vertical columns. Because of the relatively coarse resolution of the TM5-MP a-priori profiles in the standard product, TROPOMI tropospheric columns are expected to have a negative bias over polluted areas where the peak in the $NO_2$ profile is close to the surface, and where the boundary layer column is underestimated in the a-priori. Also, the time variability of the column amounts at the measurement site may be underestimated due to the a-priori. In the same way, the concentrations away from major sources may be somewhat overestimated. In Helsinki we find that replacing the original profiles with those derived from the high-resolution CAMS regional ensemble model increases the TROPOMI $NO_2$ tropospheric columns and partly reduces the discrepancy between TROPOMI and Pandora VCDs for episodes of relatively high $NO_2$ concentrations, while increasing the correlation and linear fit slope. On the other hand, the agreement does not significantly improve on average or for lower values of $NO_2$ vertical columns. Overall, the change in bias remains mostly within the uncertainties.

The overestimation of low $NO_2$ columns suggests a possible overestimation of the stratospheric fraction of the column. Also, replacing the surface reflectance climatology (Kleipool et al., 2008) currently used in the retrieval with higher resolution geometry-dependent information is expected to improve the comparison of the TROPOMI $NO_2$ vertical columns with the ground-based observations.

As compared to previous satellite-based instruments such as OMI, the bias against ground-based observations in Helsinki is similar on average (±5 % under clear sky conditions for OMI, Ialongo et al., 2016), while the correlation coefficient is generally higher for TROPOMI (r = 0.68 for TROPOMI and r = 0.5 for OMI, see Ialongo et al., 2016). The correlation between Pandora and TROPOMI $NO_2$ retrievals is also in line with the results obtained over the Canadian oil sands (r = 0.70 according to Griffin et al., 2019). On the other hand, Griffin et al. (2019) report a mean negative bias up to −30 %, as expected for very polluted sites, while we find a smaller positive bias (on average about 10 %) over a relatively less polluted site like Helsinki.

Overall, the analysis of TROPOMI $NO_2$ observations in the Helsinki area shows high correlation with ground-based observations, as well as demonstrates TROPOMI's capability to properly reproduce the temporal (day-to-day and weekly) variability of the surface $NO_2$ concentrations. This is a confirmation that satellite-based observations can bring additional information on the temporal and spatial variability of $NO_2$ in the neighbourhood of major cities, in addition to traditional air quality measurements.

*Data availability.* The re-processed TROPOMI data before 30 April 2018 were downloaded from the Sentinel-5P Expert Users Data Hub (https://s5pexp.copernicus.eu/dhus, no open access) as part of the S5P validation team activities, and after that date from the S5P Pre-Operations Data Hub (https://s5phub.copernicus.eu/dhus, open access). Pandora #105 total column data belong to the Pandonia network and are available at . Surface concentration data at the Kumpula air quality station were obtained from the FMI measurement database (no open access); an alternative source is the SmartSMEAR service (https://avaa.tdata.fi/web/smart, open access). CAMS regional forecasts and analyses for the previous day, as well as CAMS global forecasts are available through Copernicus Atmosphere Monitoring Service data portal (https://atmosphere.copernicus.eu/data). The wind data are part of the ECMWF ERA5 reanalysis product and were downloaded from the Climate Data Store (https://cds.climate.copernicus.eu).

*Author contributions.* I.I. and H.V. designed the content of the paper ans carried on the data analysis. E.H. and J.D. provided their expertise on TROPOMI NO2 retrievals and provided the CAMS model calculations. J.H. was responsible for the Pandora data. All the authors participated in writing the manuscripts.

*Competing interests.* No competing interests are present.

5  *Acknowledgements.* The research activity has been supported by the EU Horizon 2020 project E-Shape (grant agreement n.820852), by the Academy of Finland (Project ILMApilot, n.303876) and by ESA EO Science for Society funding scheme (Project DACES).

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
