# Peer review of "Comparison of TROPOMI/Sentinel 5 Precursor NO2 observations with ground-based measurements in Helsinki"

_Atmospheric Measurement Techniques, 2019_

## Referee Comment (RC1) · Anonymous Referee #1 · 1 Oct 2019

The manuscript by lolango et al. compares satellite-borne TROPOMI NO2 measurements to ground-based PANDORA NO2 measurements in Helsinki. This paper is very well written (I don't have many minor comments) and contributes to the TROPOMI validation effort. The topic of the manuscript is important as currently not many validation papers have been published and it is important to validate the measurements taken by the new satellite instrument TROPOMI with ground-based observations. In terms of methods, there is not much new added in this paper and it is actually quite similar to lolango et al. (2016) except that it uses TROPOMI measurements instead of OMI measurements. I have a few suggestions how the manuscript could be strengthen. My greatest concern is their method of how the high-resolution model CAMS was used to

re-estimate the TROPOMI tropospheric columns.

I would advise some major revisions mostly concerning the re-calculated tropospheric VCDs. The manuscript can be published in Atmospheric Measurement Techniques after these issues have been addressed.

**Major concerns**

1) lolango et al. claim that a-priori profiles have been replaced with high-resolution CAMS profiles (e.g. in the abstract p.1 l. 7; p.4 l. 25-27; p.14 l. 3-5). However, this is not true when reading the method section (p.7 l. 1-7); in fact, the tropospheric columns are simply scaled with the tropospheric CAMS columns (not profiles). Replacing the a priori profile shape with the profile shape of a high resolution model is a common technique to improve satellite tropospheric NO2 columns. However, to do this new AMF have to be estimated, e.g. Goldberg et al. (2019), McLinden et al. (2014); Russell et al. (2011), Palmer et al. (2001); Martin et al. (2002) and lots more. The a priori vertical column densities do not have a linear relation to the TROPOMI tropospheric columns. To replace the standard low resolution profile shape with that from a high resolution regional model, an new AMF has to be estimated; the relationship is not simple due to the radiative transfer in the atmosphere.

In the comparison, it can be seen that this is not a good method as the columns are simply scaled, leading to a worse product than the standard tropospheric columns. As the CAMS model is a high resolution model near a city or hot spot, these columns will be larger than for the lower resolution TM5-MP model, leading to R>1 (in eq. 3), and thus all TROPOMI columns are scaled up. Thus, it is intuitive that the scaled columns are better for high concentrations, but overall worse.

I would suggest to either use CAMS to estimate new AMFs (similar to the references provided above), or to cut this part out of the manuscript.

If CAMS is used to estimate the AMF, more description of the model is needed, from

the description on p.4 I.25-30 it is not clear what time stamp was used. Is an hourly output used? Are these interpolated to the time of the overpass?

I am also confused, why CAMS above 3km was used (3-5km). The largest impact on the tropospheric AMF comes from the high concentrations near the surface (in the boundary layer) around cities or other NOx sources. High-resolution models are used to improve the satellite tropospheric columns, because of the improved profile shape primarily in the boundary layer close to the emission sources, not to correct for the profile shape of the free-troposphere.

2) The Kumpula AQ in situ measurements are converted from surface concentrations to total columns, based on the correlation between the PANDORA and in situ measurements. One concern is that these two instruments are not co-located and are quite likely measuring two different airmasses. Especially, since the in situ measurements are taken near an airport, and thus have likely high concentrations near the surface that may or may not be captured by PANDORA, depending on the winds etc. Further, the good correlation is primarily driven by three measurements that measured high amounts of NO2 for the PANDORA and in situ measurements. I would suggest cutting this figure (Fig. 5), since it is not used for any qualitative comparison, a similar figure is provided in Fig. 2.

3) A little more can be done in this paper in terms of validation. Here are some suggestions:

- On p.4, I.1-3 lolongo et al. claim that the differences should be small between the OFFL and NRTI version. I think this paper would provide a good opportunity to quantitatively identify the differences between the NO2 NRTI and OFFL version (e.g. similar as Garane et al., 2019 who quantified the differences between the OFFL and NRTI TROPOMI O3 columns to ground-based observations).

- There may be limited measurements available but perhaps looking at the differences between TROPOMI and PANDORA NO2 columns in terms of TROPOMI's SZA, cloud

СЗ

fraction etc. similar as in Beak et al. (2017) Fig. 5 or Fig. 7

- Further, adding a boxplot showing the differences between the TROPOMI and PAN-DORA columns binned in low, medium, high columns (e.g. 0-0.6, 0.6-1, >1 10^16 molec/cm2) would also improve the paper and provide more contents to the discussion. This is already discussed on p.10 I.1-5, but a figure would help.

- The paper would improve if the time period of the comparison could be increased maybe use 1 year of data (April 2018 to April 2019). Maybe one concern would be data in the winter time with snow cover, but the difference between summer and winter observations could also be investigated.

Minor comments

Figure 2: The lines are confusing and misleading, the columns are completely unknown when no measurements are taken. I would suggest replacing the line plot with a scatter plot, at the very least for the TROPOMI, and PANDORA 10min avg. measurements.

Figure 3: It's hard to tell the difference between weekdays and weekends. I would suggest replacing the "weekend marker" with a triangle marker (or something similar). It is also sufficient to reduce the size to a 1-column plot.

P. 2 I. 5: "Netherlands" -> "Netherlands Space Office"

p. 3 I. 10: According to the AMT author guidelines dates should be written as dd month year: "on the 13th October" -> "on 13 October"

p.3 I. 14: "UV-Visible (UVVIS)" -> "UV-VIS" (as defined on p.2 I. 24)

p.3 I. 20 DOAS already defined on p.2 I. 25

p.3 I. 29: "15.04-30.09.2018" -> "15 April to 30 September 2018"

p. 3 l. 32, p.4. l. 1 : NRT -> NRTI

p.4 l. 12 : 15.04.2018-30.09.2018 -> 15 April to 30 September 2019

p.4. I. 18 -21: maybe move Fig. S1 from the supplement into the main paper. It is discussed here in a few sentences and seems important.

p.6 I. 3: FMI not defined, please define. Also, are these ground-based measurements publically available? If, so please provide the link where it can be downloaded.

p. 10 I. 11: Figure S2 -> Fig. S2 (from AMT author guidelines)

p.10 I. 25-30: as suggested in the previous section, this can be cut together with Fig.5

p. 13 I. 22: "We find this partially..." -> this has not been concluded or found from the analysis in this paper; maybe change it to : "This is partly due to the profile shapes of the low resolution TM5-MP model used to compute the standard TROPOMI tropospheric NO2 columns and thus..."

p. 15 mention that this study is using summer observations only (unless the time period has been changed, see previous suggestions), with no snow cover (?)

p.15 l. 4: the comparison to the results from Griffin et al. could be a bit more quantitatively: were the results similar, how similar? Include some numbers.

References

Baek, K. et al.: Validation of Brewer and Pandora measurements using OMI total ozone, Atmospheric Environment, Volume 160, 2017, Pages 165-175, https://doi.org/10.1016/j.atmosenv.2017.03.034.

Garane, K., et al: TROPOMI/S5ptotal ozone column data: global ground-based validation & consistency with other satellite missions, Atmos. Meas. Tech. Discuss., https://doi.org/10.5194/amt-2019-147, in review, 2019.

Goldberg, D. L., et al.: A top-down assessment using OMI NO2 suggests an underestimate in the NOx emissions inventory in Seoul, South Korea, during KORUS-AQ, Atmos. Chem. Phys., 19, 1801–1818, https://doi.org/10.5194/acp-19-1801-2019, 2019.

Martin, R. V., et al.: An improved retrieval of tropospheric nitrogen dioxide from GOME, J. Geophys. Res., 107, 4437, doi:10.1029/2001JD001027, 2002.

McLinden, C. A., et al.: Improved satellite retrievals of NO2 and SO2 over the Canadian oil sands and comparisons with surface measurements, Atmos. Chem. Phys., 14, 3637-3656, https://doi.org/10.5194/acp-14-3637-2014, 2014.

Palmer, P. I., et al.: Air mass factor formulation for spectroscopic measurements from satellites: Application to formaldehyde retrievals from the Global Ozone Monitoring Experiment, J. Geophys. Res., 106, 14539–14550, 2001.

Russell, A. R., et al.: A high spatial resolution retrieval of NO2 column densities from OMI: method and evaluation, Atmos. Chem. Phys., 11, 8543–8554, https://doi.org/10.5194/acp-11-8543-2011, 2011.

---

## Referee Comment (RC2) · Anonymous Referee #2 · 2 Oct 2019

This paper presents a comparison between satellite-based TROPOMI NO2 products and ground-based Pandora observations in Helsinki. The validation results show TROPOMI's applicability for monitoring pollution levels in urban sites, even in a relatively small and high-latitude city. I recommend publishing the paper after minor revision.

General comments:

1. The validation is based on total columns. The reason for doing so is reasonable for me. However, we usually rely on tropospheric columns to investigate air pollution. I would recommend adding the analysis focus on tropospheric columns, even though

systematic retrieval errors may exist. Such validation results will be very useful for data users to have a better sense about the current quality of the data.

2. The comparison with OMI. The authors have performed a similar validation of OMI NO2 columns against Pandora observation. Do the validation results differ significantly from this study? I would recommend a short discussion to compare the OMI and TROPOMI validations.

3. The use of high-resolution profile. I expect a better performance of the NO2 products using CAMS profiles compared to those using TM5 profiles based on the experience on OMI validations. However, as shown on Page 13, the use of CAMS a-priori profiles does not improve the agreement with Pandora significantly. What is the most likely reason for this? Does it indicate that TM5 profiles are good enough for the retrieval?

Specific comments:

1. Page 3, line 1. "The improved resolution of TROPOMI retrievals is expected to reduce the effect of dilution, due to the relatively coarse pixel size as compared to the field-of-view of the ground-based observations." I guess the authors want to say the pixel size of TROPOMI is finer than that of OMI and thus the effect of dilution is reduced. If so, what the reason for pointing out the relatively coarse pixel size as compared to the field-of-view of the ground-based observations here?

2. Page 3, line 29. The time format of "15.4.–30.9.2018" is a little bit confusing for readers. I recommend using the April 15- Sep 30. Same comments for Page 4, line 30.

3. Page 12, line 4. The authors use summed columns for TROPOMI and total columns for Pandora. Is this intended? If so, please clarify the reason in the text.

4. Page 15. Line 4. "The correlation between Pandora and TROPOMI NO2 retrievals is also in line with the results obtained by Griffin et al. (2019) over the Canadian oil sands." How those two studies are in line with each other? I recommend presenting

AMTD
the quantitative analysis for the consistency.

---

## Referee Comment (RC3) · Steven Compernolle (Referee) · 4 Oct 2019

I read the article of lalongo et al., discussing comparisons of S5p TROPOMI NO2 with ground-based data, with great interest. Given the significance of S5p TROPOMI NO2 for air quality assessments across the globe, its validation is clearly of great importance. I recommend publication in AMT with minor revisions.

**Overall.**

1/ There are indicators for bias (the MD and MRD) but not for the dispersion of differences, for example the standard deviation of the differences or the interquartile range

of the differences. Please add e.g., the standard deviation of the differences to the methodology, together with the definitions for MD and MRD, and discuss the results in the manuscript, including table 1 and 2.

2/ Although the uncertainties of S5p NO2 (p. 4) and Pandora (p. 5) are shortly mentioned, it is not discussed (e.g., in the conclusions) whether discrepancies between S5p and Pandonia are reasonable with respect to the uncertainties. Both S5p NO2 and Pandora measurements have an uncertainty provided per measurement. In the time series of co-located points of S5p NO2 and Pandora, the error bars based on the provided uncertainties can be added. It can then also be discussed whether the S5p values based on the CAMS a-priori are meaningfully different from the TM5-MP based S5p values.

3/ Minor comment: be consistent in the units for NO2 column number density, and preferably use  $10^{15}$  molec cm-2 as unit in the Tables and figures, as this is very commonly used in NO2 column comparisons. Currently the authors use  $10^{14}$  molec cm-2 in table 1 and 2, and  $10^{16}$  molec cm-2 in e.g., Fig. 5.

**Detailed comments**

Abstract, line 5. 'TROPOMI total columns underestimate ground-based observations for relatively large Pandora NO2 total columns'. It should be added here that TROPOMI overestimates for the lower columns. Also the obtained bias (absolute scale and relative), and the dispersion of the differences (e.g., the standard deviation of differences, as noted above) should be added in the abstract.

Abstract, line 9. Here it is stated that "Replacing the coarse a-priori NO2 profiles with high-resolution profiles from the CAMS chemical transport model improves the agreement between TROPOMI and Pandora total columns for episodes of NO2 enhancement." Please add a statement on the overall agreement and/or episodes of low NO2.

Introduction. p. 2, around line 27. Here, the authors should add that there is an operational validation of S5p products by the S5P-MPC-VDAF (S5P - Mission Performance Center - Validation Analysis Facility, http://mpc-vdaf.tropomi.eu/) which includes online comparisons and validation reports using the S5p total NO2 vs Pandora from the Pandonia Global Network, including the one at the Helsinki site.

p. 4, line 4. I would add here that the summed total column is the one that is recommended by the data provider.

p. 4, line 27 and following. More detail should be provided here:

- Is reanalysis data used ?
- make clear that CAMS global, despite the name similarity, is a very different model compared to CAMS regional
- add reference for CAMS global, the horizontal resolution, and the vertical range.
- 'better description of free troposphere': do you mean better compared to TM5-MP
  ?
- make more clear that you are actually constructing a hybrid profile from CAMS regional and CAMS global.
- line 29. '...using the CAMS (...) a-priori profiles'. Certainly this first time, I suggest to formulate instead 'using the hybrid CAMS regional/CAMS global a-priori profiles (called shorthand "CAMS a-priori profile" from now on) ' or some similar formulation.
- line 30. 'These ratios were available on the regular CAMS 0.1x0.1 grid' This sounds as if the authors obtained the AMF ratios from elsewhere. But if I understood well, you actually calculated the ratios yourself, using input from the hybrid CAMS regional/CAMS global profile and from the S5p product, right? Also, the

procedure how to calculate the AMF ratio using CAMS a priori data and S5p NO2 input (averaging kernel, TM5-based AMF) should be explained. E.g., likely there was need for (i) a vertical regridding of the CAMS profile to match the vertical grid of the averaging kernel of S5p NO2, and (ii) an horizontal interpolation (if so, what kind of interpolation) of the CAMS global profile to the CAMS regional grid.

These details can be discussed here, or alternatively in an appendix or the supplement.

p. 6, line 20. 'Pandora retrievals with data quality flag value of 0, 1, 10 or 11'. Pandora measurements can occasionally become negative and even reach several Pmolec cm-2 in the negative. This is drastically reduced when only focusing on high-quality data with 0, 10 flags. Was there any filtering on negative Pandora values, or were these averaged together with the positive values, or were these -by chance- no longer present after co-location with TROPOMI?

p. 7, fig. 2. I share the concerns of reviewer 1 on the clarity of this figure.

p. 7, line 5. 'CAMS a priori summed column' is somewhat ambiguous. A reader could assume this is a column purely derived from CAMS information. I suggest: 'the newly derived summed column, using the CAMS a-priori profile,...,is calculated as...'

p. 7, line 2. 'ratio (R) between the tropospheric column retrievals...' This is unclear. From section 2.1, I assume R is the ratio of the original  $AMF_{trop}$  of the S5p NO2 product and the newly calculated  $AMF_{trop}$ .

p. 7, Eq (3). From the formula, it is clear that the stratospheric contribution is not updated (still based on TM5-MP), while CAMS global is nonetheless available (as the authors used it for the free troposphere). A motivation is needed why CAMS regional+global is used for the troposphere while TM5 is kept for the stratosphere.

p. 9, Table 1.

· Regarding the slope from orthogonal regression, it should be noted in the text

that this technique assumes that the standard deviation from random error in y (S5p NO2 total column) and x (Pandora total column) are equal, which is not at all guaranteed. See e.g., Carroll (1996), with  $\eta$  of Eq (4) assumed 1, or Wu (2018), who do not recommend orthogonal distance regression.

• What is the meaning of the number after the  $\pm$  ? Is it the standard deviation of the mean? This should be explained in the table footnote. Similar for Table 2.

p. 10, line 19. What is the impact of changing the co-location criteria (spatial and temporal) on the standard deviation of the differences and the correlation coefficient?

p. 10, line 23. What is meant by 'variability' here? The amount by which the MD changes?

p. 12, Fig. 5 right panel. Add error bars (based on the provided uncertainties) to S5p NO2 and Pandonia points. This figure will be clearer when using points instead of lines.

p. 12, Fig. 6. What is the meaning of the vertical error bars? The standard deviation of the values in the month? This should be explained in the caption.

p. 12-13 ( about the evaluation of the effect of using CAMS a-priori profiles) + Fig. S3

- Please add in Fig. S3 error bars on the S5p NO2 TM5-MP points and on the Pandonia points. This will give an indication whether the update with the CAMS a-priori profiles is significant with respect to the uncertainties.
- Assumed that the numbers after the  $\pm$  in Table 2 are standard deviations of the mean, it seems to me that the difference between the MD calculated with TM5-MP profiles on the one hand, and the MD calculated with CAMS a-priori on the other hand, is not statistically significant. Same remark for the MRD. This should then be also reflected in the abstract and the conclusions.

p. 13 line 4-5. 'On the other hand, in cases of low concentrations, where TROPOMI tends to overestimate the VCDs compared to Pandora, the use of CAMS a-priori profiles slightly worsens the agreement with Pandora by increasing the positive bias. ' Looking at Fig S3 this effect seems really small to me and is probably not statistically significant. Add in Table 2 entries for 'Pandora high' and 'Pandora low' so one can conclude what is the significance of this effect.

p. 13, Conclusions. Here, it should also be stated whether the S5p vs Pandora discrepancies are reasonable (or not) in light of the measurement uncertainties of S5p and Pandora.

p. 13, line 22. 'while low values are overestimated' A short discussion on the possible reasons should go here. Does this mean that TROPOMI has a positive systematic error at low NO2 values? Or that the Pandora instrument has a negative systematic error? Or is it somehow due to the still relatively coarse resolution of S5p NO2? And is the overestimation actually significant with respect to the uncertainties?

p. 15, Data availability. It should be noted that there is no general open access to the S5p Expert users Data Hub, only to the S5p Pre-Operations Data Hub. Also, the point of access for CAMS regional and CAMS global should added here, and exactly which kind of data was used (forecast, reanalysis?).

**References**

Carroll, R. J. and Ruppert, D. The Use and Misuse of Orthogonal Regression in Linear Errors-in-Variables Models The American Statistician, 1, feb 1996, 50

Wu, C. and Yu, J. Z. Evaluation of linear regression techniques for atmospheric applications: the importance of appropriate weighting Atmos. Meas. Tech., 2, 2018, 11, 1233-1250

---

## Author Comment (AC1) · 29 Nov 2019

**We thank the referee 1 for the comments and we answer to the specific questions below. The referee's comments are in black while the answers by the authors are in blue.**

1) Ialongo et al. claim that a-priori profiles have been replaced with high-resolution CAMS profiles (e.g. in the abstract p.1 l. 7: p.4 l. 25-27: p.14 l. 3-5). However, this is not true when reading the method section (p.7 l. 1-7); in fact, the tropospheric columns are simply scaled with the tropospheric CAMS columns (not profiles). Replacing the a priori profile shape with the profile shape of a high resolution model is a common technique to improve satellite tropospheric NO2 columns. However, to do this new AMF have to be estimated, e.g. Goldberg et al. (2019), McLinden et al. (2014); Russell et al. (2011), Palmer et al. (2001); Martin et al. (2002) and lots more. The a priori vertical column densities do not have a linear relation to the TROPOMI tropospheric columns. To replace the standard low resolution profile shape with that from a high resolution regional model, an new AMF has to be estimated; the relationship is not simple due to the radiative transfer in the atmosphere. In the comparison, it can be seen that this is not a good method as the columns are simply scaled, leading to a worse product than the standard tropospheric columns. As the CAMS model is a high resolution model near a city or hot spot, these columns will be larger than for the lower resolution TM5-MP model, leading to R>1 (in eq. 3), and thus all TROPOMI columns are scaled up. Thus, it is intuitive that the scaled

columns are better for high concentrations, but overall worse.

I would suggest to either use CAMS to estimate new AMFs (similar to the references provided above), or to cut this part out of the manuscript.

If CAMS is used to estimate the AMF, more description of the model is needed, from the description on p.4 l.25-30 it is not clear what time stamp was used. Is an hourly output used? Are these interpolated to the time of the overpass?

I am also confused, why CAMS above 3km was used (3-5km). The largest impact on the tropospheric AMF comes from the high concentrations near the surface (in the boundary layer) around cities or other NOx sources. High-resolution models are used to improve the satellite tropospheric columns, because of the improved profile shape primarily in the boundary layer close to the emission sources, not to correct for the profile shape of the free-troposphere.

The approach to replace the a-priori is very briefly described on page 4, line 22-30, by referring to the Product User Manual (PUM) where the procedure is described. This approach provides a new estimate of the tropospheric column by using the full profile to recompute the air-mass factor. On page 7 we describe how the total column comparison is made, by updating only the troposphere (new a-priori) and keeping the stratosphere unchanged (eq. 3).

The response of the referee made us realise that the explanation how this is done was too short. Indeed the reader may get the impression that we simply use ratios of tropospheric columns. This is not the case and we clarify this in sect. 2.3 as well. As mentioned, the recipe to replace the a-priori is described by Eskes et al. (2019). This approach makes use of the averaging kernels and involves integrals over the

profiles, so the full profile shape is used. This new profile shape leads to a new AMF. As mentioned by the referee, there is no direct relation between the a-priori column and the retrieved column, since only the profile shape determines the AMF. This approach, based on the averaging kernels, works if only the a-priori profile of NO2 is replaced and no other inputs for the retrieval are changed. The approach makes use of the fact that NO2 is optically thin (which is valid except for incidental extremely high tropospheric columns).

The referee mentions several papers where the air-mass factors were recomputed. For instance, McLinden et al. (2014), but also the POMINO product over China (Lin, J. T. et al., 2014) introduce high-resolution regional model outputs to improve the retrievals on a regional scale, similar to what is presented in our paper. However, in these papers not only the a-priori is replaced, but also other aspects of the retrieval are modified, such as the use of alternative (high-resolution) albedo maps or the explicit treatment of aerosols. In these cases, indeed, the radiative transfer calculation has to be done again to compute the impact on the tropospheric air-mass factor, because these changes also lead to a change of the averaging kernels. The approach described in the PUM no longer works.

The averaging kernels in case of clear and weakly clouded scenes decreases when moving from the tropopause down to the surface. Indeed, as mentioned by the referee the column amount above 3 km is small compared to the column amount in the boundary layer. But, because the sensitivity in the free troposphere is much higher (e.g. factor of 3 is normal) we find that this small free troposphere column still has a substantial impact on the AMF especially in the more rural areas. The regional models are not designed to describe the free troposphere accurately and produce unrealistically low NO2 above 2-3 km. This is why we combined profile information from the CAMS-global system (3 km to tropopause) with the CAMSregional profiles below 3 km.

To explain this also in the paper, we extended the last paragraph of section 2.1 (page 4):

"Since the retrieval of TROPOMI vertical column densities (VCDs) is sensitive to the a-priori estimate of the NO2 profile shape, the accuracy of the VCDs may be improved by using a-priori profiles from a chemical transport model (CTM) with a higher resolution than the 1°×1° of TM5-MP (Williams et al., 2017). The air-mass factor (AMF) can be recomputed using an alternative a-priori NO2 profile, resulting in a new retrieval of the tropospheric NO2 column as described by Eskes et al. (2019).

In order to analyse their impact on the comparison, below 3 km altitude we used NO2 profiles from the CAMS regional ENSEMBLE model (Météo-France, 2016; Marécal et al., 2015) as an alternative to the TM5-MP profiles. The CAMS regional ENSEMBLE is a median of seven European CTMs, and the data are provided on a regular 0.1°×0.1° grid over Europe on 8 vertical levels up to 5 km altitude. In

addition, the CAMS global model was used to generate the profiles above 3 km altitude with the assumption that this model gives a more reliable description of NOx in the free troposphere. Data for CAMS global are provided on a regular  $0.4^{\circ} \times 0.4^{\circ}$  grid on 60 model levels reaching up to 0.1 hPa (Flemming et al., 2015). In particular, we used the ratios between TROPOMI tropospheric air-mass factors derived using the hybrid CAMS regional/global a-priori profile (henceforth "CAMS a-priori") and the TM5-MP a-priori profile (see Sect 2.3). These ratios were provided on the regular CAMS  $0.1^{\circ} \times 0.1^{\circ}$  grid for the period 30 April to 30 September 2018.

In order to minimize representativeness errors during the comparison, certain considerations were taken into account so that the fields could be correctly sampled in space and time. Horizontally, all available gridded data were interpolated to the CAMS regional, 0.1°×0.1° grid. Source grids in this process were either the TROPOMI native grid, which is different for each orbit, the CAMS global grid or the TM5-MP grid. Horizontal interpolation of retrieval columns was realized by means of a weighted average of all individual columns within a target grid cell. Intensive variables (e.g. temperatures, pressures, averaging kernels, the tropopause layer index etc.) were interpolated horizontally using bilinear regridding. Modelled fields were also interpolated in time, based on the satellite overpass time over Central Europe. All vertical levels of source data were linearly interpolated to the TM5-MP vertical levels and all subsequent integrations to columns were performed based on those levels. Pressures at each of those levels were calculated based on the surface pressure and the hybrid coefficients included in the TROPOMI product, which originate in TM5-MP. For the column integrations, all concentrations were converted to densities based on temperature and pressure profiles provided by TM5-MP."

2) The Kumpula AQ in situ measurements are converted from surface concentrations to total columns, based on the correlation between the PANDORA and in situ measurements. One concern is that these two instruments are not colocated and are quite likely measuring two different airmasses. Especially, since the in situ measurements are taken near an airport, and thus have likely high concentrations near the surface that may or may not be captured by PANDORA, depending on the winds etc. Further, the good correlation is primarily driven by three measurements that measured high amounts of NO2 for the PANDORA and in situ measurements. I would suggest cutting this figure (Fig. 5), since it is not used for any qualitative comparison, a similar figure is provided in Fig. 2.

This was a misunderstanding. The AQ station and Pandora are indeed co-located. They are about 100 meters from each other in the Kumpula area of Helsinki. The confusion came perhaps from the two points in Fig. 1. We clarify this in the text. We find figure 5 important to visualize the temporal correspondence between in situ measurements and satellite observations; we remove now the lines to make it clearer as suggested by the referee n.3. We add this sentence in section 2.2: "This station, also known as SMEAR III station (Järvi et al., 2009), is located close to the Pandora instrument (about 100 m distance)."

3) A little more can be done in this paper in terms of validation. Here are some suggestions:

3a) On p.4, l.1-3 Ialongo et al. claim that the differences should be small between the OFFL and NRTI version. I think this paper would provide a good opportunity to quantitatively identify the differences between the NO2 NRTI and OFFL version (e.g. similar as Garane et al., 2019 who quantified the differences between the OFFL and NRTI TROPOMI O3 columns to ground-based observations).

The NRTI data are not stored and are replaced with the OFFL in the sentinel data hub, so the NRTI data are not available for a comparison in the past. Nevertheless, there is an operational validation of S5p products by the S5P-MPC-VDAF (S5P -Mission Performance Center - Validation Data Analysis Facility, http://mpcvdaf.tropomi.eu/), which includes online comparisons between both NRTI and OFFL NO2 products and the Pandora NO2 total columns from the Pandonia Global Network, including the Helsinki site. The results are summarized in 3-montly validation reports and they show almost identical results (see for example the last report here: http://mpc-vdaf.tropomi.eu/ProjectDir/reports/pdf/S5P-MPC-IASB-ROCVR-04.0.0-20190923\_FINAL.pdf)

We add this document as reference to the text and we mention the operational validation activities as also suggested by referee n.3.

3b) There may be limited measurements available but perhaps looking at the differences between TROPOMI and PANDORA NO2 columns in terms of TROPOMI's SZA, cloud fraction etc. similar as in Beak et al. (2017) Fig. 5 or Fig. 7

We add plots in the supplement including the bias vs SZA and CRF but we note that we apply already a screening to the data that removes cloudy pixels and high SZA values. There is an apparent increase in bias (first positive, then negative) with increasing CRF but less clear with SZA. We also analyse the bias vs the time of the day and pixel number and we update the text as follows:

"Figure S6 in the Supplement includes the absolute differences between TROPOMI and Pandora NO2 total columns as a function of TROPOMI SZA (solar zenith angle) and CRF (cloud radiative fraction) (upper and lower panel, respectively) within the range of values allowed after the TROPOMI data screening (QA value >0.75). While the dependence between the differences and SZA values is not clear, the differences for SZA above 45° are generally larger (between -3 and +1e15 molec./cm2) than for smaller SZA values (0 to 1e15 molec./cm2). Similarly, larger CRF values correspond to larger (positive or negative) absolute differences.

Since S5P has often two valid overpasses per day at the latitude of Helsinki 60°N), it is possible to study the NO2 daily variability between about 12 and 15LT. The S5P overpass time typically corresponds to the NO2 daily local minimum (between the morning and afternoon peaks due to commuter traffic), observed for example in the NO2 surface concentration measurements from Kumpula AQ site (Fig. S7). Figure 5 (upper panel) shows TROPOMI and Pandora NO2 total columns as a function of the time of the day between 12 and 15 LT. Both datasets show an enhancement around 13:30LT and lower NO2 levels before and after. The relative differences between TROPOMI and Pandora NO2 total columns do not show a clear dependence on the time of the day (Fig. 5, lower panel), but the dispersion (standard deviation of the relative differences) is larger (about 30%) before 13:30 LT than afterwards (21%). Increasing time of the day also corresponds to increasing pixel number (filled colour dots in Fig. 5, lower panel), since the first overpass of the day corresponds to the left side of the orbit (smaller pixel numbers) while the second overpass to the right side (higher pixels number). No clear dependence between the relative differences and the pixel size (larger at the edges and smaller in the center of the swath) was observed."

Further, adding a boxplot showing the differences between the TROPOMI and PANDORA columns binned in low, medium, high columns (e.g. 0-0.6, 0.6-1, >1 1016 molec/cm2) would also improve the paper and provide more contents to the discussion. This is already discussed on p.10 l.1-5, but a figure would help.

**We added a box plot in the supplement as suggested.**

- The paper would improve if the time period of the comparison could be increased maybe use 1 year of data (April 2018 to April 2019). Maybe one concern would be data in the winter time with snow cover, but the difference between summer and winter observations could also be investigated.

This is unfortunately not possible because we have no measurements from Pandora for winter or for year 2019 due to maintenance. TROPOMI data also are not available at Helsinki latitude for more than a couple of months in winter, after the quality flag screening. Further analysis will be perhaps the focus of a future work, when a larger amount of data are collected.

**Minor comments**

Figure 2: The lines are confusing and misleading, the columns are completely unknown when no measurements are taken. I would suggest replacing the line plot with a scatter plot, at the very least for the TROPOMI, and PANDORA 10min avg. measurements.

We changed figure 2 according to the suggestions.

Figure 3: It's hard to tell the difference between weekdays and weekends. I would suggest replacing the "weekend marker" with a triangle marker (or something similar). It is also sufficient to reduce the size to a 1-column plot.

We changed figure 3 according to the suggestions.

P. 2 l. 5: "Netherlands" -> "Netherlands Space Office"

Changed

p. 3 l. 10: According to the AMT author guidelines dates should be written as dd month year: "on the 13th October" -> "on 13 October"

**Changed**

p.3 l. 14: "UV-Visible (UVVIS)" -> "UV-VIS" (as defined on p.2 l. 24)

**Changed**

p.3 l. 20 DOAS already defined on p.2 l. 25

**Removed**

p.3 l. 29: "15.04-30.09.2018" -> "15 April to 30 September 2018"

**Changed**

p. 3l. 32,p.4. l. 1: NRT->NRTI

**Changed**

p.4 l. 12 : 15.04.2018-30.09.2018 -> 15 April to 30 September 2019

**Changed**

p.4. l. 18 -21: maybe move Fig. S1 from the supplement into the main paper. It is discussed here in a few sentences and seems important.

We think that the supplement is more appropriate for such technical maps.

p.6 l. 3: FMI not defined, please define. Also, are these ground-based measurements publically available? If, so please provide the link where it can be downloaded.

**Changed**

p. 10 l. 11: Figure S2 -> Fig. S2 (from AMT author guidelines)

**Changed**

p.10 l. 25-30: as suggested in the previous section, this can be cut together with Fig.5

We leave it together with the picture.

p. 13 l. 22: "We find this partially. . ." -> this has not been concluded or found from the analysis in this paper; maybe change it to : "This is partly due to the profile shapes of the low resolution TM5-MP model used to compute the standard TROPOMI tropospheric NO2 columns and thus. . ."

We change the sentence as: "This is partly due to the low resolution of the TM5-MP profile shapes used to compute the tropospheric air-mass factors and thus the vertical columns."

p. 15 mention that this study is using summer observations only (unless the time period has been changed, see previous suggestions), with no snow cover (?)

**Added**

p.15 l. 4: the comparison to the results from Griffin et al. could be a bit more quantitatively: were the results similar, how similar? Include some numbers.

We refer now to the correlation coefficient and bias values as follows: "The correlation between Pandora and TROPOMI NO2 retrievals is also in line with the results obtained over the Canadian oil sands (r=0.70 according to Griffin et al., 2019). On the other hand, Griffin et al. (2019) report a mean negative bias up to -30%, as expected for very polluted sites, while we find a smaller positive bias (on average about 10%) over a relatively less polluted site like Helsinki."

---

## Author Comment (AC2) · 29 Nov 2019

**We thank reviewer 2 for the comments and we answer to the specific questions below. The referee's comments are in black while the answers by the authors are in blue.**

**General comments:**

1. The validation is based on total columns. The reason for doing so is reasonable for me. However, we usually rely on tropospheric columns to investigate air pollution. I would recommend adding the analysis focus on tropospheric columns, even though systematic retrieval errors may exist. Such validation results will be very useful for data users to have a better sense about the current quality of the data.

We validate summed columns as they are those comparable with the ground-based Pandora observations. We do not have equivalent measurements of tropospheric NO2 from a ground-based instrument in Helsinki. On the other hand, we use the tropospheric columns for qualitative analysis as the weekly cycle, for example.

2. The comparison with OMI. The authors have performed a similar validation of OMI NO2 columns against Pandora observation. Do the validation results differ significantly from this study? I would recommend a short discussion to compare the OMI and TROPOMI validations.

We mentioned this but we write now in more details in the conclusion as follows: "As compared to previous satellite-based instruments such as OMI, the bias against ground-based observations in Helsinki is similar on average ( $\pm 5\%$  under clear sky conditions for OMI, Ialongo et al. (2016)), while the correlation coefficient is generally higher (r=0.68 for TROPOMI and r=0.5 for OMI, see Ialongo et al., 2016)."

3. The use of high-resolution profile. I expect a better performance of the NO2 products using CAMS profiles compared to those using TM5 profiles based on the experience on OMI validations. However, as shown on Page 13, the use of CAMS a-priori profiles does not improve the agreement with Pandora significantly. What is the most likely reason for this? Does it indicate that TM5 profiles are good enough for the retrieval?

Indeed the improvement is not significant on average but it is sensible for episodes with high NO2 columns as measured by Pandora. The improvement is expected to improve the retrieval under polluted conditions where the spatial variability is sharper, but we have in Helsinki also several overpass with somewhat background conditions, so that the change overall remains small (within the uncertainties). Also, Griffin et al. 2019 also stated that using high-resolution input improves the tropospheric AMF and the tropospheric NO2 VCDs but the correction is not as significant as previously seen for OMI. That study included also a better characterization of snow-covered surfaces.

We update the text in the Sect. Results as follows:

"The comparison shows that the largest differences between the two summed columns are mostly found in cases of relatively high concentrations. In these cases, the use of CAMS profiles generally increases the TROPOMI summed columns and reduces the difference between TROPOMI and Pandora (from  $-28.5\pm3.3$  % for TM5-MP to  $-23.7\pm3.5$  % for CAMS). On the other hand, in cases of low concentrations, where TROPOMI tends to overestimate the VCDs compared to Pandora, the use of CAMS a-priori profiles slightly increases the positive bias (from  $+16.9\pm2.3$  % for TM5-MP to  $+19.1\pm2.3$  % for CAMS). Because the largest improvement is achieved for relatively high concentrations and negative biases becoming less negative, the overall MRD value increases from 11.5 % to 14 % (Table 2). According to a two-sided t-test, the differences of the two mean absolute biases (MD) in Table 2 are statistically significant at the 52% significance level. Thus, on average, the use of CAMS profiles does not improve significantly the agreement with Pandora observations.

For this smaller subset of 75 co-locations with Pandora the correlation between TM5-MP summed columns and Pandora is 0.74 and the slope of a least squares linear fit is 0.45. Using the CAMS profiles improves the agreement with Pandora in terms of correlation and slope, with their values increasing to 0.80 and 0.52, respectively. This improvement is more evident for high values of the Pandora NO2 total columns with the correlation and the linear slope increasing by 0.1 and 0.27, respectively, from TM5-MP to CAMS (Table 2).

The time series in Fig. S8 of the supplement further illustrate how using the highresolution CAMS profiles increases the TROPOMI tropospheric columns so that the summed columns (yellow dots) become closer to Pandora's peak values (blue dots), corresponding to episodes of NO2 enhancement, but that overall the difference between the summed columns obtained using TM5-MP and CAMS remains mostly within the uncertainties of the TROPOMI NO2 retrieval."

We clarify this also in the abstract and conclusion, respectively, as follows:

**Abstract:**

"Replacing the coarse a-priori NO2 profiles with high-resolution profiles from the CAMS chemical transport model improves the agreement between TROPOMI and Pandora total columns for episodes of NO2 enhancement. When only the low values of NO2 total columns or the whole dataset are taken into account, the mean bias slightly increases. The change in bias remains mostly within the uncertainties."

**Conclusion:**

"In Helsinki we find that replacing the original profiles with those derived from the high-resolution CAMS regional ensemble model increases the TROPOMI NO2 tropospheric columns and partly reduces the discrepancy between TROPOMI and Pandora VCDs for episodes of relatively high NO2 concentrations, while increasing the correlation and the linear fit slope. On the other hand, the agreement does not significantly improve on average or for lower values of NO2 vertical columns. Overall, the change in bias remains mostly within the uncertainties."

Specific comments:

1. Page 3, line 1. "The improved resolution of TROPOMI retrievals is expected to reduce the effect of dilution, due to the relatively coarse pixel size as compared to the field-of-view of the ground-based observations." I guess the authors want to say the pixel size of TROPOMI is finer than that of OMI and thus the effect of dilution is reduced. If so, what the reason for pointing out the relatively coarse pixel size as compared to the field-of-view of the ground-based observations here?

We mean here that the smaller pixels of TROPOMI (compared to OMI) will possibly reduce the dilution effect when compared to the field-of-view of the ground-based observations.

We rewrite this as: "The improved resolution of TROPOMI retrievals is expected to reduce the effect of spatial averaging compared to OMI, leading to a better agreement with the ground-based Pandora observations that has a relatively narrow field-of-view."

2. Page 3, line 29. The time format of "15.4.–30.9.2018" is a little bit confusing for readers. I recommend using the April 15- Sep 30. Same comments for Page 4, line 30.

We changed that throughout the manuscript according to the recommendations for AMT journal

3. Page 12, line 4. The authors use summed columns for TROPOMI and total columns for Pandora. Is this intended? If so, please clarify the reason in the text.

Yes it was on purpose. We explained that we used the summed over the total column product, because of the latter's sensitivity to the ratio between the stratospheric and tropospheric a-priori columns may lead to substantial systematic retrieval errors. The intermediate step of using data assimilation to first estimate the stratospheric column does remove part of this error.

We add also this sentence in the text to further clarify:

"The summed total column product is described by the data provider as the best physical estimate of the NO2 vertical column and recommended for comparison to ground-based total column observations (van Geffen et al., 2019)."

4. Page 15. Line 4. "The correlation between Pandora and TROPOMI NO2 retrievals is also in line with the results obtained by Griffin et al. (2019) over the Canadian oil sands." How those two studies are in line with each other? I recommend presenting the quantitative analysis for the consistency.

We rewrite the text as follows: "The correlation between Pandora and TROPOMI NO2 retrievals is also in line with the results obtained over the Canadian oil sands (r=0.70 according to Griffin et al., 2019). On the other hand, Griffin et al. (2019)

report a mean negative bias up to -30%, as expected for very polluted sites, while we find a smaller positive bias (on average about 10%) over a relatively less polluted site like Helsinki."

---

## Author Comment (AC3) · 29 Nov 2019

**We thank the referee S. Compernolle for the useful comments and we answer to the specific questions below. The referee's comments are in black while the answers by the authors are in blue.**

Overall

1/ There are indicators for bias (the MD and MRD) but not for the dispersion of differences, for example the standard deviation of the differences or the interquartile range of the differences. Please add e.g., the standard deviation of the differences to the methodology, together with the definitions for MD and MRD, and discuss the results in the manuscript, including table 1 and 2.

We added the SD of the differences in Table 1 and 2 and we briefly discuss it in the text.

2/ Although the uncertainties of S5p NO2 (p. 4) and Pandora (p. 5) are shortly mentioned, it is not discussed (e.g., in the conclusions) whether discrepancies between S5p and Pandonia are reasonable with respect to the uncertainties. Both S5p NO2 and Pandora measurements have an uncertainty provided per measurement. In the time series of co-located points of S5p NO2 and Pandora, the error bars based on the provided uncertainties can be added. It can then also be discussed whether the S5p values based on the CAMS a-priori are meaningfully different from the TM5-MP based S5p values.

We added the errorbars in the Fig. 2 (and Fig. S8 of the updated supplement), as suggested. We discuss now in more details how the observed discrepancies compares to the uncertainties as follows:
"We find that the differences between the total columns derived from the TROPOMI and Pandora instruments are on average around 10 % (or $0.12 \times 10^{15}$ molec./cm$^{-2}$), which is smaller than the precision of the TROPOMI summed columns used in this study (10–50%) and well below the requirements for TROPOMI observations (25–50 % for the NO2 tropospheric column and 10 % for the stratospheric column; ESA, 2017)."

We also discuss the significance of the change of a-priori as described in the following points.

3/ Minor comment: be consistent in the units for NO2 column number density, and preferably use 1015 molec cm−2 as unit in the Tables and figures, as this is very commonly used in NO2 column comparisons. Currently the authors use 1014 molec cm−2 in table 1 and 2, and 1016 molec cm−2 in e.g., Fig. 5.

All pictures and tables are corrected accordingly to this suggestion

Detailed comments

4/ Abstract, line 5. 'TROPOMI total columns underestimate ground-based observations for relatively large Pandora NO2 total columns'. It should be added here that TROPOMI overestimates for the lower columns. Also the obtained bias (absolute scale and relative), and the dispersion of the differences (e.g., the standard deviation of differences, as noted above) should be added in the abstract.

The following text was added to the abstract:
"The mean relative and absolute bias between the TROPOMI and Pandora NO2 total columns is about +10% and 0.12e15 molec./cm^2, respectively. The dispersion of these differences (estimated as their standard deviation) is 2.2e15 molec./cm^2."
[…]
"On the other hand, TROPOMI slightly overestimates (within the retrieval uncertainties) relatively small NO2 total columns."

Abstract, line 9. Here it is stated that " Replacing the coarse a-priori NO2 profiles with high-resolution profiles from the CAMS chemical transport model improves the agreement between TROPOMI and Pandora total columns for episodes of NO2 enhancement." Please add a statement on the overall agreement and/or episodes of low NO2.

We added the following text to the abstract:
"When only the low values of NO2 total columns or the whole dataset are taken into account, the mean bias slightly increases. The change in bias remains mostly within the uncertainties."

Introduction. p. 2, around line 27. Here, the authors should add that there is an operational validation of S5p products by the S5P-MPC-VDAF (S5P - Mission Performance Center - Validation Analysis Facility, http://mpc-vdaf.tropomi.eu/) which includes online comparisons and validation reports using the S5p total NO2 vs Pandora from the Pandonia Global Network, including the one at the Helsinki site.

We added this text to the introduction:
"The TROPOMI/S5P NO2 products are operationally validated by the S5P-MPC-VDAF (S5P - Mission Performance Center - Validation Data Analysis Facility) using the Pandora NO2 total columns from the PGN. The operational validation results are reported every 3 months at the S5P-MPC-VDAF website (http://mpc-vdaf.tropomi.eu/)"

p. 4, line 4. I would add here that the summed total column is the one that is recommended by the data provider.

We add this sentence in the text to further clarify:
"The summed total column product is described by the data provider as the best physical estimate of the NO2 vertical column and recommended for comparison to ground-based total column observations (van Geffen et al., 2019)."

p. 4, line 27 and following. More detail should be provided here:
• Is reanalysis data used ?
• make clear that CAMS global, despite the name similarity, is a very different model compared to CAMS regional

• add reference for CAMS global, the horizontal resolution, and the vertical range.
• 'better description of free troposphere': do you mean better compared to TM5-MP ?
• make more clear that you are actually constructing a hybrid profile from CAMS regional and CAMS global.
• line 29. '...using the CAMS (...) a-priori profiles'. Certainly this first time, I sug- gest to formulate instead 'using the hybrid CAMS regional/CAMS global a-priori profiles (called shorthand "CAMS a-priori profile" from now on) ' or some similar formulation.
• line 30. 'These ratios were available on the regular CAMS 0.1x0.1 grid' This sounds as if the authors obtained the AMF ratios from elsewhere. But if I under- stood well, you actually calculated the ratios yourself, using input from the hybrid CAMS regional/CAMS global profile and from the S5p product, right? Also, the procedure how to calculate the AMF ratio using CAMS a priori data and S5p NO2 input (averaging kernel, TM5-based AMF) should be explained. E.g., likely there was need for (i) a vertical regridding of the CAMS profile to match the vertical grid of the averaging kernel of S5p NO2, and (ii) an horizontal interpolation (if so, what kind of interpolation) of the CAMS global profile to the CAMS regional grid.

We try to answer all your questions by changing/adding the text at the end of Section 2.1 as follows:

"Since the retrieval of TROPOMI vertical column densities (VCDs) is sensitive to the a-priori estimate of the NO2 profile shape, the accuracy of the VCDs may be improved by using a-priori profiles from a chemical transport model (CTM) with a higher resolution than the 1°×1° of TM5-MP (Williams et al., 2017). The air-mass factor (AMF) can be recomputed using an alternative a-priori NO2 profile, resulting in a new retrieval of the tropospheric NO2 column as described by Eskes et al. (2019).
In order to analyse their impact on the comparison, below 3 km altitude we used NO2 profiles from the CAMS regional ENSEMBLE model (Météo-France, 2016; Marécal et al., 2015) as an alternative to the TM5-MP profiles. The CAMS regional ENSEMBLE is a median of seven European CTMs, and the data are provided on a regular 0.1°×0.1° grid over Europe on 8 vertical levels up to 5 km altitude. In addition, the CAMS global model was used to generate the profiles above 3 km altitude with the assumption that this model gives a more reliable description of NOx in the free troposphere. Data for CAMS global are provided on a regular 0.4°×0.4° grid on 60 model levels reaching up to 0.1 hPa (Flemming et al., 2015). In particular, we used the ratios between TROPOMI tropospheric air-mass factors derived using the hybrid CAMS regional/global a-priori profile (henceforth "CAMS a-priori") and the TM5-MP a-priori profile (see Sect 2.3). These ratios were provided on the regular CAMS 0.1°×0.1° grid for the period 30 April to 30 September 2018.
In order to minimize representativeness errors during the comparison, certain considerations were taken into account so that the fields could be correctly sampled in space and time. Horizontally, all available gridded data were interpolated to the CAMS regional, 0.1°×0.1° grid. Source grids in this process were either the TROPOMI native grid, which is different for each orbit, the CAMS global grid or the TM5-MP

grid. Horizontal interpolation of retrieval columns was realized by means of a weighted average of all individual columns within a target grid cell. Intensive variables (e.g. temperatures, pressures, averaging kernels, the tropopause layer index etc.) were interpolated horizontally using bilinear regridding. Modelled fields were also interpolated in time, based on the satellite overpass time over Central Europe. All vertical levels of source data were linearly interpolated to the TM5-MP vertical levels and all subsequent integrations to columns were performed based on those levels. Pressures at each of those levels were calculated based on the surface pressure and the hybrid coefficients included in the TROPOMI product, which originate in TM5-MP. For the column integrations, all concentrations were converted to densities based on temperature and pressure profiles provided by TM5-MP."

These details can be discussed here, or alternatively in an appendix or the supplement. p. 6, line 20. 'Pandora retrievals with data quality flag value of 0, 1, 10 or 11'. Pandora measurements can occasionally become negative and even reach several Pmolec cm- 2 in the negative. This is drastically reduced when only focusing on high-quality data with 0, 10 flags. Was there any filtering on negative Pandora values, or were these averaged together with the positive values, or were these -by chance- no longer present after co-location with TROPOMI?

Negative values were filtered out (they showd negative uncertainty as well) but they actually appeared only in two cases and including those in the calculation only changes the bias by a few decimals.

p. 7, fig. 2. I share the concerns of reviewer 1 on the clarity of this figure.

We changed it according to the suggestions

p. 7, line 5. 'CAMS a priori summed column' is somewhat ambiguous. A reader could assume this is a column purely derived from CAMS information. I suggest: 'the newly derived summed column, using the CAMS a-priori profile,...,is calculated as...'

We changed this with: "The new summed column, derived using the CAMS a-priori profile, was then calculated…"

p. 7, line 2. 'ratio (R) between the tropospheric column retrievals...' This is unclear. From section 2.1, I assume R is the ratio of the original AMFtrop of the S5p NO2 product and the newly calculated AMFtrop .

Yes, thank you. This was a mistake. We rewrite as follows:
"The effect of using high-resolution CAMS a-priori $NO_2$ profiles instead of TM5-MP (as used in the standard product) in the calculation of TROPOMI VCDs was analysed by calculating an alternative summed column using the ratio (R) between 5 the tropospheric air-mass factors derived using CAMS and TM5-MP a-priori profiles, computed on the CAMS-regional grid with 0.1° resolution (see Sect.

2.1).”

p. 7, Eq (3). From the formula, it is clear that the stratospheric contribution is not updated (still based on TM5-MP), while CAMS global is nonetheless available (as the authors used it for the free troposphere). A motivation is needed why CAMS regional+global is used for the troposphere while TM5 is kept for the stratosphere.

The retrieval includes an assimilation step to minimize the bias between the TM5-MP modeled and observed stratospheric column as much as possible. This is an essential element of the retrieval and should only be replaced when the other model has a high quality stratospheric NO2 and assimilates the satellite data to get a comparable or better analysis.
At this moment CAMS-global does not include detailed stratospheric chemistry, and the NO2 profiles in the stratosphere are poor. Secondly, CAMS assimilates only tropospheric columns from OMI and GOME-2 which does not impact the stratosphere.

We add this sentence: “The stratospheric columns from TM5-MP (as in the standard product) are used in the calculation of the new summed columns, because at the moment CAMS global does not include detailed stratospheric chemistry nor accurate NO2 profile information in the stratosphere.”

p. 9, Table 1.
• Regarding the slope from orthogonal regression, it should be noted in the text C4 that this technique assumes that the standard deviation from random error in y (S5p NO2 total column) and x (Pandora total column) are equal, which is not at all guaranteed. See e.g., Carroll (1996), with η of Eq (4) assumed 1, or Wu (2018), who do not recommend orthogonal distance regression.

We replace the orthogonal regression with both the least square fit slope as well as the York fit slope as recommended by Wu et al. (2018) and we add this sentence:
“The York linear regression (York et al. 2004) is used alongside the traditional least squares linear regression, since it has been shown to be an appropriate measure of fit in situations where the two sets of data have different levels of uncertainty (Wu et al., 2018).”

• What is the meaning of the number after the ± ? Is it the standard deviation of the mean? This should be explained in the table footnote. Similar for Table 2.

Yes it is. We clarify this in the captions of both tables.

p. 10, line 19. What is the impact of changing the co-location criteria (spatial and temporal) on the standard deviation of the differences and the correlation coefficient?

We add now a plot in the supplement with the correlation coefficient and the standard deviation of the differences as a function of the changing co-location criteria in the supplement and we update the text accordingly.

p. 10, line 23. What is meant by 'variability' here? The amount by which the MD changes?

This sentence is removed and replaced with: "The MD value increases with increasing temporal averaging interval by about 0.3e15molec./cm2 (2 percentage points)."

p. 12, Fig. 5 right panel. Add error bars (based on the provided uncertainties) to S5p NO2 and Pandonia points. This figure will be clearer when using points instead of lines.

Corrected

p. 12, Fig. 6. What is the meaning of the vertical error bars? The standard deviation of the values in the month? This should be explained in the caption.

Yes it is. Corrected

p. 12-13 ( about the evaluation of the effect of using CAMS a-priori profiles) + Fig. S3
• Please add in Fig. S3 error bars on the S5p NO2 TM5-MP points and on the Pandonia points. This will give an indication whether the update with the CAMS a-priori profiles is significant with respect to the uncertainties.

Corrected. Note that S3 is S8 in the revised manuscript.

• Assumed that the numbers after the ± in Table 2 are standard deviations of the mean, it seems to me that the difference between the MD calculated with TM5- MP profiles on the one hand, and the MD calculated with CAMS a-priori on the other hand, is not statistically significant. Same remark for the MRD. This should then be also reflected in the abstract and the conclusions.

Indeed the improvement is not significant on average but it is sensible for episodes with high NO2 columns as measured by Pandora. The improvement is expected to improve the retrieval under polluted conditions where the spatial variability is sharper, but we have in Helsinki also several overpass with somewhat background conditions, so that the change overall remains small (within the uncertainties).

We update the text in the Sect. Results as follows:

"The comparison shows that the largest differences between the two summed columns are mostly found in cases of relatively high concentrations. In these cases, the use of CAMS profiles generally increases the TROPOMI summed columns and reduces the difference between TROPOMI and Pandora (from -28.5±3.3 % for TM5-MP to -23.7±3.5 % for CAMS). On the other hand, in cases of low concentrations,

where TROPOMI tends to overestimate the VCDs compared to Pandora, the use of CAMS a-priori profiles slightly increases the positive bias (from +16.9±2.3 % for TM5-MP to +19.1±2.3 % for CAMS). Because the largest improvement is achieved for relatively high concentrations and negative biases becoming less negative, the overall MRD value increases from 11.5 % to 14 % (Table 2). According to a two-sided t-test, the differences of the two mean absolute biases (MD) in Table 2 are statistically significant at the 52% significance level. Thus, on average, the use of CAMS profiles does not improve significantly the agreement with Pandora observations.

For this smaller subset of 75 co-locations with Pandora the correlation between TM5-MP summed columns and Pandora is 0.74 and the slope of a least squares linear fit is 0.45. Using the CAMS profiles improves the agreement with Pandora in terms of correlation and slope, with their values increasing to 0.80 and 0.52, respectively. This improvement is more evident for high values of the Pandora NO2 total columns with the correlation and the linear slope increasing by 0.1 and 0.27, respectively, from TM5-MP to CAMS (Table 2).

The time series in Fig. S8 of the supplement further illustrate how using the high-resolution CAMS profiles increases the TROPOMI tropospheric columns so that the summed columns (yellow dots) become closer to Pandora's peak values (blue dots), corresponding to episodes of NO2 enhancement, but that overall the difference between the summed columns obtained using TM5-MP and CAMS remains mostly within the uncertainties of the TROPOMI NO2 retrieval."

We clarify this also in the abstract and conclusion, respectively, as follows:

Abstract:
 "Replacing the coarse a-priori NO2 profiles with high-resolution profiles from the CAMS chemical transport model improves the agreement between TROPOMI and Pandora total columns for episodes of NO2 enhancement. When only the low values of NO2 total columns or the whole dataset are taken into account, the mean bias slightly increases. The change in bias remains mostly within the uncertainties."

Conclusion:
"In Helsinki we find that replacing the original profiles with those derived from the high-resolution CAMS regional ensemble model increases the TROPOMI NO2 tropospheric columns and partly reduces the discrepancy between TROPOMI and Pandora VCDs for episodes of relatively high NO2 concentrations, while increasing the correlation and the linear fit slope. On the other hand, the agreement does not significantly improve on average or for lower values of NO2 vertical columns. Overall, the change in bias remains mostly within the uncertainties."

p. 13 line 4-5. 'On the other hand, in cases of low concentrations, where TROPOMI tends to overestimate the VCDs compared to Pandora, the use of CAMS a-priori pro- files slightly worsens the agreement with Pandora by increasing the positive bias. ' Looking at Fig S3 this effect seems really small to me and is probably not statistically significant.

Add in Table 2 entries for 'Pandora high' and 'Pandora low' so one can conclude what is the significance of this effect.

We updated table 2 accordingly. See also the answer to the previous point.

p. 13, Conclusions. Here, it should also be stated whether the S5p vs Pandora discrepancies are reasonable (or not) in light of the measurement uncertainties of S5p and Pandora.

Corrected as follows:
"We find that the differences between the total columns derived from the TROPOMI and Pandora instruments are on average around 10 % (or $0.12 \times 10^{15}$ molec. $cm^{-2}$), which is smaller than the precision of the TROPOMI summed columns used in this study (10–50 %) and well below the requirements for TROPOMI observations (25–50 % for the $NO_2$ tropospheric column and <10 % for the stratospheric column; ESA, 2017)."

p. 13, line 22. 'while low values are overestimated' A short discussion on the possible reasons should go here. Does this mean that TROPOMI has a positive systematic error at low NO2 values? Or that the Pandora instrument has a negative systematic error? Or is it somehow due to the still relatively coarse resolution of S5p NO2? And is the overestimation actually significant with respect to the uncertainties?

The overestimation of low NO2 columns suggests a possible overestimation of the stratospheric fraction of the column. Also, replacing the surface reflectivity climatology (Kleipool et al., 2008) currently used in the retrieval with higher resolution geometry-dependent information is expected to improve the comparison of the TROPOMI NO2 vertical columns with the ground-based observations.
Anyway, the reasons for this positive bias are still under investigation. We mention this in the text.

p. 15, Data availability. It should be noted that there is no general open access to the S5p Expert users Data Hub, only to the S5p Pre-Operations Data Hub. Also, the point of access for CAMS regional and CAMS global should added here, and exactly which kind of data was used (forecast, reanalysis?).

We correct that and we add this text:
"CAMS regional forecasts and analyses for the previous day, as well as CAMS global forecasts are available through Copernicus Atmosphere Monitoring Service data portal (https://atmosphere.copernicus.eu/data)."